# Opponent Modeling based on Subgoal Inference

**Xiaopeng Yu**    **Jiechuan Jiang**    **Zongqing Lu**[†]

School of Computer Science, Peking University

## Abstract

When an agent is in a multi-agent environment, it may face previously unseen opponents, and it is a challenge to cooperate with other agents to accomplish the task together or to maximize its own rewards. Most opponent modeling methods deal with the non-stationarity caused by unknown opponent policies via predicting the opponent's actions. However, focusing on the opponent's action is shortsighted, which also constrains the adaptability to unknown opponents in complex tasks. In this paper, we propose *opponent modeling based on subgoal inference*, which infers the opponent's subgoals through historical trajectories. As subgoals are likely to be shared by different opponent policies, predicting subgoals can yield better generalization to unknown opponents. Additionally, we design two subgoal selection modes for cooperative games and general-sum games respectively. Empirically, we show that our method achieves more effective adaptation than existing methods in a variety of tasks.

## 1 Introduction

Autonomous agents are systems capable of making decisions and acting independently in their environment, often operating without direct human intervention [3]. These agents can either cooperate with or compete against each other, depending on the context. In cooperative scenarios, many multi-agent reinforcement learning (MARL) methods [20, 41, 33, 38] aim to bridge the information gap between agents [44] by training agents in a centralized manner, called centralized training with decentralized execution, enabling agents to work together seamlessly to accomplish cooperative tasks. Alternatively, fully decentralized methods[17, 40] seek to break free from the constraints of centralized training, allowing agents to reach collaboration in a simpler and decentralized manner. In competitive scenarios, NFSP [15], PSRO [19], and Deep-

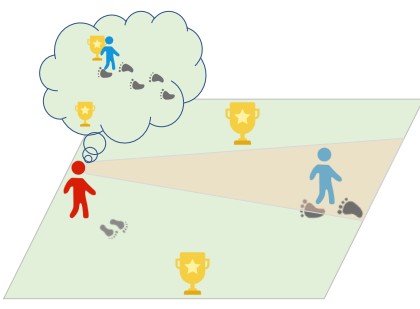

Figure 1: Infer the goal of others

Nash [29] employ self-play to train agents for equilibrium strategies, allowing agents to adapt and improve their policy. By considering how the agent affects the expected learning progress of other agents, LOLA [10] and COLA [49] apply opponent shaping to this setting. Overall, these methods focus on training agents in a way that accounts for their interactions, resulting in a set of policies that enable effective collaboration or competition within a group of agents.

While the above methods emphasizes the collective behavior of agents, it is also crucial to consider the role of individual agents, particularly self-interested agents, in these multi-agent environments. A self-interested agent [37, 12] operates with the primary goal of maximizing its own benefits,

---

[†]Correspondence to ✉ zongqing.lu@pku.edu.cn

38th Conference on Neural Information Processing Systems (NeurIPS 2024).

even when interacting with other agents. When the objectives of a self-interested agent align with those of the team, this scenario falls under ad-hoc teamwork [21, 8, 39]; however, in more general cases, these interactions are framed as noncooperative games [34, 23, 45]. A key technique for self-interested agents in such settings is opponent modeling[24, 3], which enables them to analyze and predict the actions, goals, and beliefs of other agents. By modeling the intentions and policies of other agents, the training process of the agent might be stabilized [27]. Many studies rely on predicting the actions [14, 16, 13, 25, 26], goals [32, 31], and returns [42] of opponents during training. Then, the autonomous agent can adapt to different or unseen opponents by using the predictions or representations that are produced by the relevant modules.

Although a lot of the existing methods concentrate on modeling the opponent's actions, we argue that such an approach is short-sighted, pedantical, and highly complex. Generally, modeling an opponent's actions just predicts what it will do at the next step. Intuitively, it is more beneficial for the agent to make decisions if it knows the situation of the opponent several steps ahead. Predicting the actions over a few steps has high uncertainty. For example, to reach the goal point of $(2, 2)$, an opponent moves from $(0, 0)$ following the action sequence $<\uparrow, \uparrow, \rightarrow, \rightarrow>$ by four steps (Cartesian coordinates). But, there are also 5 other action sequences, *i.e.,* $<\uparrow, \rightarrow, \uparrow, \rightarrow>, <\uparrow, \rightarrow, \rightarrow, \uparrow>, <\rightarrow, \uparrow, \uparrow, \rightarrow>, <\rightarrow, \uparrow, \rightarrow, \uparrow>, <\rightarrow, \rightarrow, \uparrow, \uparrow>$, that can lead to the same goal. Obviously, the complexity and uncertainty of predicting the action sequence are much higher than the goal itself. Other methods that claim to predict the opponent's goal [31, 32], but without explicitly making a connection to the opponent's goal or just predicting the goal at the next step, are essentially as shortsighted as modeling actions.

Inspired by the fact that humans can predict the opponent's goal by observing the opponent's actions for several steps as illustrated in Figure 1, in this paper, we propose ***O**pponent **M**odeling based on sub**G**oals inference* (**OMG**), which uses variational inference to predict the opponent's future subgoals from historical trajectories. The trajectory of an opponent's policy consists of a set of subgoals, and the trajectories of different policies may contain the same subgoals. This combinatorial property of the subgoals facilitates the generalization of the agent to unseen opponents' policies. Moreover, we design two manners for selecting subgoals, which are applied to cooperative games and general sum games, respectively. Empirically, OMG outperforms existing opponent modeling methods in a variety of multi-agent environments, demonstrating the superiority of inferring subgoals over predicting actions.

## 2 Related Work

**Opponent modeling.** Opponent modeling plays a crucial role in enhancing the robustness and stability of reinforcement learning [27]. Given the presence of diverse opponent policies in multi-agent environments, the autonomous agent faces a significant challenge in learning resilient policies. When an agent perceives an opponent as part of the environment, the resulting environment becomes inherently unstable and intricate. To address this challenge, one straightforward method involves equipping the agent with the ability to incorporate information about its opponent, including aspects like the opponent's behavior, goals, and beliefs [3], *i.e.,* opponent modeling. It gives the agent a deeper insight and prediction ability about the opponent's policy. Thus, the autonomous agent views the environment as less unstable and can simply use single-agent reinforcement learning methods.

A common approach to modeling the policy of an opponent is predicting the opponent's actions. DRON [14] and DPIQN [16] extend DQN [22] by adding another network that estimates the opponents' actions from the observations. The DQN uses the hidden layer of this network to improve its policy. Variational auto-encoders can also be used to model the opponent's policy [25], which results in probabilistic representations instead of fixed vectors. PR2 [48] and TP-MCTS [47] combine the idea of recursive reasoning, nested form as "the agent believes [that the opponent believes (that the agent believes ...)]", based on modeling the action of the opponent. Some works focus on modeling beliefs. [53] combined the sequential and hierarchical variational auto-encoders to construct a belief inference model using meta-learning, for belief inference. [51] introduced landmarks into the behavior model and improved the model by the action sequence of the opponents, so as to recognize and compare the opponent's intention.

Another key aspect of opponent modeling is to infer the opponent's goal. [5] formulated the goal recognition as a Markov decision process (MDP) and calculated the posterior probability of the

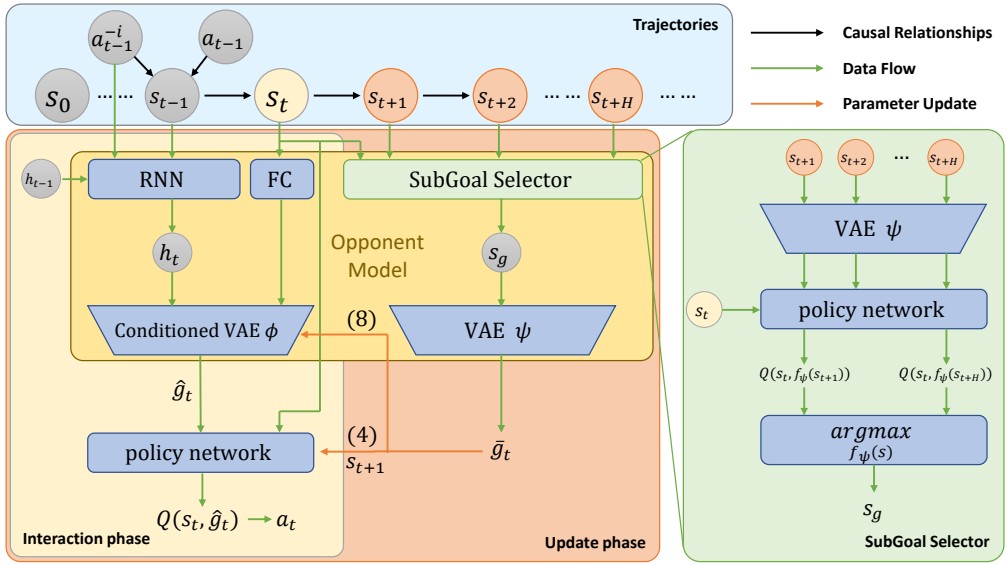

Figure 2: Diagram of OMG. During the interaction phase, OMG infers the subgoal from the historical trajectory. The CVAE $\phi$ acts as an inference model, deducing the opponent subgoal denoted as $\hat{g}$. The inferred subgoal serves as input for the policy model $Q(s, \hat{g}, a)$. In the update phase, OMG examines the entire trajectory in a hindsight manner to select subgoals $\bar{g}$ as priors for training the inference model. The subgoal selector employs a value-based heuristic to choose a state from the next few steps and then encodes it into a subgoal using the pre-trained VAE $\psi$.

goal by Bayes' rule based on a prior goal library. ToMnet [31] aims to give the agent a human-like Theory of Mind. It uses three networks to infer the agent's goal and action from previous and present information. SOM [32] implements the Theory of Mind with a goal library from a different perspective. SOM uses its own policy, the opponent's observation, and the opponent's action to work backward to learn the opponent's goal distribution by gradient ascent. These methods either require a prior goal library or infer implicit "goals" that are not supervised by ground truth goals.

In some scenarios, opponents may continuously learn during interaction. Meta-MAPG [18] combines Meta-PG [1] and LOLA [10], and focuses on the problem of the non-stationary environment caused by the continuous learning of opponents. MBOM [50] simultaneously targets a variety of adversaries, fixed policy, or continuous learning, by modeling the possible policies that an opponent may form, combined with Bayesian inference to generate an opponent's imagined policy. GSCU [11] chooses online between a real-time greedy strategy and a fixed conservative strategy through Bayesian belief in competitive environments. Unlike these methods, in this paper, we consider the most common setting where opponents have unseen, diverse, but fixed policies during test.

**Goal-conditioned RL.** Goal-conditioned reinforcement learning is an extension of the single-agent algorithm. Most works focus on learning a goal-conditioned policy, where the goals are usually predefined [30, 52]. Some works consider acquiring subgoals automatically to accelerate learning. [28] proposed a method that uses expert trajectories to generate subgoals, while [7] proposed to incorporate imaginary subgoals into policy learning to facilitate learning complex tasks, where subgoals are measured by value functions. Unlike existing goal-conditioned RL methods, we aim to infer the subgoal of the opponent and condition the agent policy on the inferred subgoal.

## 3 Preliminaries

In general, we consider an $n$-agent stochastic game $\mathcal{M} = (\mathcal{S}, \mathcal{A}^1, \ldots, \mathcal{A}^n, \mathcal{P}, \mathcal{R}^1, \ldots, \mathcal{R}^n, \gamma)$, where $\mathcal{S}$ is the state space, $\mathcal{A}^i$ is the action space of agent $i \in [1, \ldots, n]$, $\mathcal{A} = \prod_{i=1}^{n} \mathcal{A}^i$ is the joint action space of agents, $\mathcal{P} : \mathcal{S} \times \mathcal{A} \times \mathcal{S} \to [0, 1]$ is a transition function, $\mathcal{R}^i : \mathcal{S} \times \mathcal{A} \to \mathbb{R}$ is the reward function of agent $i$, and $\gamma \in [0, 1)$ is the discount factor. The policy of agent $i$ is $\pi^i$, and the joint policy of other agents is $\pi^{-i}(a^{-i}|s) = \prod_{j \neq i} \pi^j(a^j|s)$, where $a^{-i}$ is the joint action except agent

$i$. All agents interact with the environment simultaneously without communication. The historical trajectory is available, *i.e.*, for agent $i$ at timestep $t$, $\tau_t = \{s_0, a_0^i, a_0^{-i}, \ldots, s_{t-1}, a_{t-1}^i, a_{t-1}^{-i}\}$ is observable. The goal of the agent $i$ is to maximize its expected cumulative discount rewards:

$$\mathbb{E}_{\substack{s_{t+1} \sim \mathcal{P}(\cdot|s_t, a_t^i, a_t^{-i}), \\ a \sim \pi^i(\cdot|s_t), a_t^{-i} \sim \pi^{-i}(\cdot|s_t)}} \left[ \sum_{t=0}^{\infty} \gamma^t \mathcal{R}^i(s_t, a_t^i, a_t^{-i}) \right]. \tag{1}$$

For convenience, the learning agent treats all other agents as a joint opponent with the joint action $a^{-i} \sim \pi^{-i}(\cdot|s)$ and reward $r^{-i}$. The action and reward of the learning agent are respectively denoted as $a \sim \pi(\cdot|s)$ and $r$ for notation simplicity.

If an agent treats other agents as part of the environment and ignores the non-stationarity posed by the change of other agents' policies as independent Q-learning [43, 44]. Its Q-function $\mathcal{Q}$ is updated by:

$$\mathcal{Q}(s_t, a_t) = \mathbb{E}_{\mathcal{P}(s_{t+1}|s_t, a^{-i}, a)}[r + \gamma \max_a \mathcal{Q}(s_{t+1}, a)]. \tag{2}$$

Opponent modeling typically predicts the actions of other agents to address the non-stationary problem. The opponent model uses historical trajectory as input to predict $\tilde{a}^{-i} \sim \tilde{\pi}(\cdot|\tau)$, where $\tilde{a}^{-i}$ is the estimate of $a^{-i}$. Then, its Q-function is updated as:

$$\mathcal{Q}(s_t, \tilde{a}_t^{-i}, a_t) = \mathbb{E}_{\mathcal{P}(s_{t+1}|s_t, a^{-i}, a)}[r + \gamma \max_a \mathcal{Q}(s_{t+1}, \tilde{a}_{t+1}^{-i}, a)]. \tag{3}$$

Note that we cast our discussion here to Q-learning. All can be similarly applied to other RL methods, such as PPO [36].

## 4 Method

In this section, we present our method, opponent modeling based on subgoals inference (OMG). First, we discuss learning policies with the opponent's subgoals, compared to learning based on the opponent's actions. Then, we introduce our opponent model that infers the opponent's subgoals using a value-based heuristic.

### 4.1 Policy Learning with Opponent's Subgoals

In Equation (3), the traditional opponent modeling with the opponent's actions is introduced. Here, we introduce policy learning with the opponent's subgoals.

*The opponent's subgoals offer a more structured representation compared to individual actions.* Subgoals represent feature embeddings of future states that the opponent aims to achieve based on its policy. Although diverse action sequences can lead to the same state, focusing on subgoals provides a higher-level understanding of the opponent's long-term intentions. Instead of gaining new information, subgoal modeling reinterprets observed data to emphasize long-term objectives, reducing variability and improving learning efficiency [24]. By concentrating on the opponent's desired states rather than individual actions, the agent can achieve more stable and effective policy learning.

The opponent's subgoal distribution is determined by the opponent's action sequence, *i.e.*, the opponent's policy, but the subgoal space is still the representation of the state space. Here we decouple the subgoal from the opponent's policy and just consider decision-making problems conditioned on the opponent's subgoal. Formally, we transform the original stochastic game $\mathcal{M}$ into a state-augmented MDP, defined by $\mathcal{M}_\mathcal{G} = (\mathcal{S}, \mathcal{G}, \mathcal{A}^i, \mathcal{P}, \mathcal{R}^i, \gamma)$, where $\mathcal{G}$ is the subgoal space. $\mathcal{G}$ is a representation of future states the opponent may go, $|\mathcal{G}| \leq |\mathcal{S}|$.

The state-augmented MDP's state space $\mathcal{S}$ extends to the MDP with state-subgoal pairs $(\mathcal{S}, \mathcal{G})$. Therefore, the agent's Q-function based on the opponent's subgoal is updated as:

$$\mathcal{Q}(s_t, g_t, a_t) = \mathbb{E}_{\mathcal{P}(s_{t+1}|s_t, a^{-i}, a)}[r + \gamma \max_a \mathcal{Q}(s_{t+1}, g_t, a)]. \tag{4}$$

Here the pair $(s_{t+1}, g_t)$ is used instead of $(s_{t+1}, g_{t+1})$, as we assume that the next state of $(s_t, g_t)$ follows the same goal. In the framework of OMG, $g_t$ and $g_{t+1}$ will reach the same state at the end of the episode.

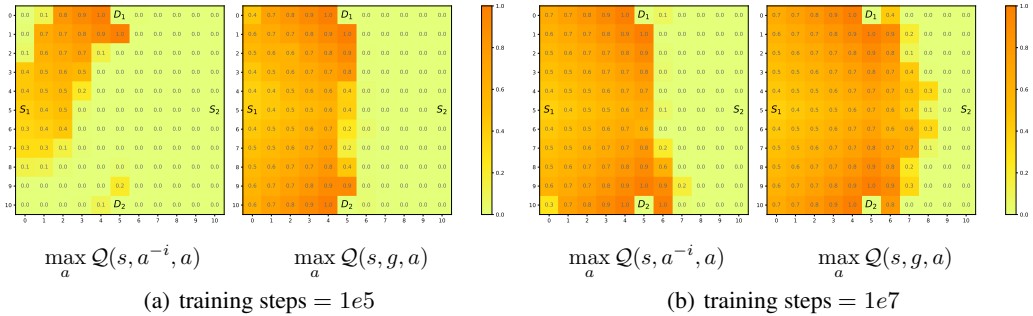

$$\max_a \mathcal{Q}(s, a^{-i}, a) \qquad \max_a \mathcal{Q}(s, g, a) \qquad \max_a \mathcal{Q}(s, a^{-i}, a) \qquad \max_a \mathcal{Q}(s, g, a)$$

(a) training steps $= 1e5$ $\qquad\qquad\qquad$ (b) training steps $= 1e7$

Figure 3: Learned Q-values using tabular Q-learning in an $11 \times 11$ gridworld. The agent and the opponent start from the $S_1$ and $S_2$, respectively. The two rewarding grids are $D_1$ and $D_2$, and the reward will only be given to the agent who arrives first. The opponent executes one of policies $\pi_1^{-i}$ and $\pi_2^{-i}$, which target $D_1$ and $D_2$, respectively. The $g$ and $a^{-i}$ are obtained from an oracle.

*Q-values with opponent's subgoals are just as effective as with opponent's actions.* We carry out an experiment in an $11 \times 11$ gridworld with two agents, as detailed in Figure 3. After convergence, the Q-value increases as the agent gets closer to the rewarding grid, indicating a meaningful Q-value with the opponent's subgoal, as shown in Figure 3(b).

*Effective opponent's subgoals enhance policy learning.* The Q-value using the opponent's action learns slower than the Q-value with the opponent's subgoal in Figure 3(a), resulting from the tuple $(s, a^{-i}, a)$ is more numerous than $(s, g, a)$ in the Q-table. When there are fewer $(s, g, a)$ than $(s, a^{-i}, a)$, the method using $(s, g, a)$ naturally holds the advantage of faster learning than the method of $(s, a^{-i}, a)$. The quantity of $(s, g, a)$ is contingent upon the goal selection, and we present an analysis of the quantitative relationship between pair $(s, g)$ and $(s, a^{-i})$, see Appendix A. In short, the number of $(s, g)$ is significantly smaller than that of $(s, a^{-i})$ in our method. Next, we explain how the opponent model secures these benefits.

### 4.2   Opponent Modeling Based on Subgoal Inference

Our opponent modeling consists of two components: *subgoal inference model* and *subgoal selector*. The subgoal inference model employs the historical trajectory to predict the opponent's subgoal, serving as input for the policy during the interaction phase. The subgoal selector scrutinizes the entire historical trajectory using a value-based heuristic to choose the appropriate subgoal for training the inference model during the update phase.

**Subgoal inference model.** The subgoal $g$ represents a feature embedding of a future state. Specifically, for a trajectory $\{s_0, a_0, a_0^{-i}, \ldots, s_t, a_t, a_t^{-i}, \ldots, s_T\}$, the state corresponding to subgoal $g_t$ at $s_t$ is one of future states $\mathcal{N}_t = \{s_{t+1}, s_{t+2}, \ldots, s_T\}$, denoted as $s_t^g$ and determined by the subgoal selector.

The objective of the subgoal inference model is to infer $g_t$ from the historical trajectory $\tau_t = \{s_0, a_0, a_0^{-i}, \ldots, s_{t-1}, a_{t-1}, a_{t-1}^{-i}\}$. This aligns with the intuitive hypothesis that the opponent's intention can often be inferred after just a few initial actions.

Here, we introduce variational inference and employ a conditional variational auto-encoder (CVAE) as the subgoal inference model. In this model, we represent the subgoal posterior probability as $q_\phi(\hat{g}_t | \tau_t, s_t)$ and the likelihood estimate as $p_\theta(s_t | \hat{g}_t, \tau_t)$ with $\phi$ and $\theta$ respectively denoting the network parameters. The subgoal prior model, denoted as $p_\psi$, is a pre-trained variational autoencoder (VAE) using the states previously collected in the environment, and produces the subgoal prior $p_\psi(\bar{g}_t | s_t^g)$ given the subgoal state $s_t^g$ chosen by the subgoal selector.

Further details about the network architecture are provided in Figure 2. The optimization objective of the subgoal inference model is:

$$< \hat{\theta}, \hat{\phi} >= \arg\max_{\theta, \phi} \mathbb{E}_{q_\phi(\hat{g}_t | \tau_t, s_t)} \Big[ \log p_\theta(s_t | \hat{g}_t, \tau_t) \Big] - \mathrm{KL}\Big( q_\phi(\hat{g}_t | \tau_t, s_t) \| p_\psi(\bar{g}_t | s_t^g) \Big). \qquad (5)$$

---

**Algorithm 1** OMG

---

1: **_Preparation:_**
2: Interact with $\nu$ opponents to collect $s$ and train the prior model $p_\psi$
3: Initialize subgoal inference model parameters $\phi$ and $\theta$
4: Initialize Q-network $\mathcal{Q}$ and the replay buffer $\mathcal{D}$
5: **repeat**
6:    **_Interaction phase_**
7:    Observe state $s$
8:    Infer $\hat{g}$ by subgoal inference model $q_\phi(\hat{g}|\tau, s)$
9:    Choose action $a$ by $\max_a \mathcal{Q}(s, \hat{g}, a)$ with $\epsilon$-greedy
10:   Store experience $(s, a, a^{-i}, r)$ in replay buffer $\mathcal{D}$
11:   **_Update phase_**
12:   **if** time to update **then**
13:      Obtain prior subgoal $\bar{g}$ by (6) or (7)
14:      Calculate subgoal $g$ by (8)
15:      Update Q-network by (4)
16:      Update subgoal inference model $q_\phi$ and $p_\theta$ by (5)
17:   **end if**
18: **until** convergence

---

where the term $p_\psi(\bar{g}|s^g)$ in the KL divergence accounts for the prior distribution and is pre-trained. The purpose of including the KL divergence term is to prevent collapse of the inference model.

**Subgoal selector.** The primary objective of the subgoal selector is to choose the appropriate future state of the subgoal state $s_t^g$ from $\mathcal{N}_t$ as input to the prior model. The choice of the subgoal state plays a significant role in shaping the agent's behavior and leaning towards either optimism or conservatism. This is especially critical when dealing with cooperative games and general-sum games, where the dynamics of interactions are complex and multifaceted. In these contexts, we provide two distinct manners for the subgoal selection:

$$\bar{g}_t = \arg\max_{s_i \in \mathcal{N}_t^H} \mathbb{E}_{g \sim p_\psi(\cdot|s_i)} V(s_t, g) \tag{6}$$

$$\bar{g}_t = \arg\min_{s_i \in \mathcal{N}_t^H} \mathbb{E}_{g \sim p_\psi(\cdot|s_i)} V(s_t, g), \tag{7}$$

where $V(s, g) = \mathbb{E}_a Q(s, g, a)$, $\mathcal{N}_t^H$ is the set of future states $\{s_{t+1}, \cdots, s_{t+H}\}$. As discussed in Section 4.1, the quantity of $(s, g)$ pairs is crucial. Selecting candidate subgoal states is pivotal in this regard. Thus, we use states within the next $H$ timesteps instead of all future states. The choice of $H$ gives a tradeoff between the agent's generalization to diverse opponents induced by the fact that the subgoals of different trajectory fragments have combinatorial properties and the learning difficulty incurred by the increased opponent subgoals.

As indicated in Equation (6), we pinpoint the subgoal within an $H$-horizon that maximizes the V-value. The agent incorporates this to optimize the Q-function, thus adopting an optimistic strategy akin to the maximax strategy [6], which applies to cooperative games. Conversely, if we choose the subgoal as in Equation (7), it corresponds to the subgoal yielding the lowest value. The agent then employs this for learning Q-function, leading to a conservative strategy similar to the minimax strategy, which is commonly used in general-sum games.

The subgoal selector and the subgoal inference model as a whole constitute our opponent modeling module. During the interaction phase, the subgoal inference model is used to get the inferred subgoal $\hat{g}$, which is combined with the state as the input to the Q-network. During the update phase, the prior subgoal $\bar{g}$ generated by the subgoal selector is provided to the inference model for training. The subgoal inference model is unstable at the beginning, which may disturb the updating of the Q-network. Therefore, we use the following combination of the prior subgoal $\bar{g}$ and the inferred subgoal $\hat{g}$ as the input of Q-network,

$$g_t = \hat{g}_t \mathbb{I}(\eta > \epsilon) + \bar{g}_t \mathbb{I}(\eta \leq \epsilon), \quad \eta \sim U[0,1], \tag{8}$$

where $\epsilon$ is a hyperparameter that decreases to zero over training. We will further empirically study this in Section 5.4.

For completeness, the full procedure of OMG is given in Algorithm 1.

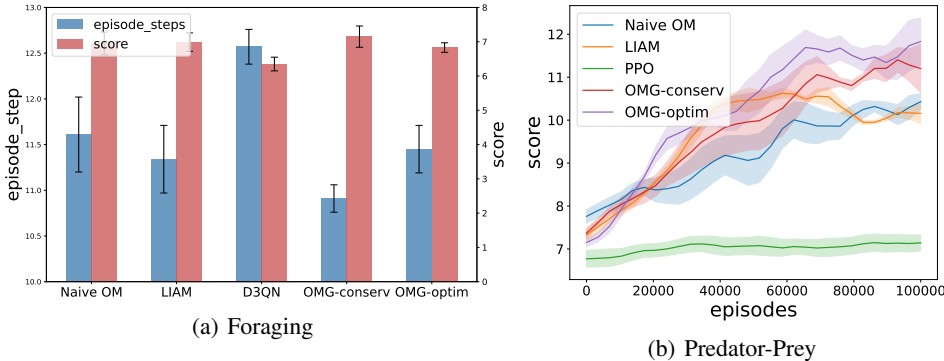

(a) Foraging

(b) Predator-Prey

Figure 4: (a) Performance in Foraging. The red bar shows the total score obtained by the agent. The blue bar illustrates the number of steps in each episode. The results show that OMG can converge to the same score as the baselines but end the episode in fewer steps because it predicts the opponent's goal. (b) Performance in Predator-Prey. The results show the score obtained by the agent as a predator with two other uncontrolled predators, and OMG outperforms the baselines.

## 5 Experiments

First, we evaluate OMG's training performance in two environments (discrete and continuous state spaces) and then test its generalization to opponents with unseen policies in a more complex environment. In all the experiments, the baselines have the same neural network architectures as OMG. All the methods are trained for five runs with different random seeds, and results are presented using mean and standard deviation. More details about experimental settings and hyperparameters are available in Appendix B. To ensure reproducibility, we include the code in the supplementary material and will make it open-source upon acceptance.

We experiment in the following three multi-agent environments. Foraging [2, 4] is an $8 \times 8$ gridworld where the agent aims to collect foods. Predator-Prey [20] is a three-against-one scenario with continuous space where the agent collaborates with predators to capture prey. SMAC [35] is a high-dimensional environment for collaborative multi-agent reinforcement learning based on StarCraft II, where the agent cooperates with a set of opponents with unknown policies to accomplish tasks.

### 5.1 Baselines

In the experiments, we implement two variants of OMG, OMG-optimistic and OMG-conservative, based on the subgoal selection manners in Equation (6) and Equation (7), respectively. OMG compared with the following methods:

- Naïve OM [14] uses observation to directly model the opponent's policy, which assists the agent in decision-making by predicting the opponent's actions.
- LIAM [26] uses the observations and actions of the opponent with an encoder-decoder architecture, and the model learns to extract representations about the opponent, conditioned only on the local observations of the controlled agent.
- D3QN & PPO & IQL [46, 36, 43] are classical RL algorithms without opponent modeling.

We use D3QN, PPO, and IQL as the backbone algorithms in Foraging, Predator-Prey, and SMAC, respectively, to reproduce the performance of baselines. The versions of OMG that are based on D3QN and IQL incorporate "dueling" and "double" tricks over Algorithm 1. For OMG based on PPO, please refer to Appendix F for details.

### 5.2 Performance of Training

We evaluate the performance of OMG on Foraging and Predator-Prey, and the results are shown in Figure 4(a) and Figure 4(b), respectively. In the foraging environment, our method attains similar scores to the baseline methods, and both the agent and the opponent achieve comparable scores.

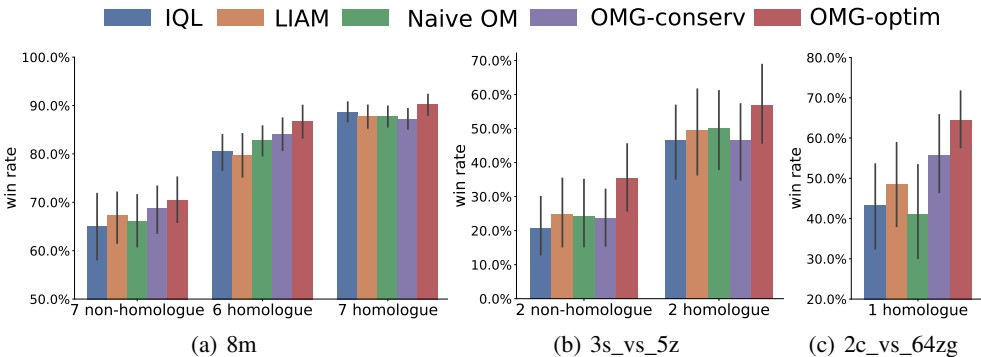

(a) 8m  (b) 3s_vs_5z  (c) 2c_vs_64zg

Figure 5: Test performance of cooperation with unseen opponents in *8m* (a), *3s_vs_5z* (b) and *2c_vs_64zg* (c) maps of SMAC. The X-axis represents the opponent's policies, and "homologue" refers to the policy learned by the same algorithm, while "non-homologue" represents different ones; e.g. 7 homologue refers to 7 opponents from 8 agents trained by the same algorithm (QMIX, VDN or IQL) on the 8m, and 7 non-homologue involves 7 opponents from different runs of those algorithms. The results show that OMG-optimistic outperforms all baselines. The results are averaged over collaborating with 30 opponents of different policies, with 95% confidence intervals.

OMG has a shorter episode length compared to other methods because OMG can predict the subgoal that the opponent is heading to and thus avoid wasting steps in the same direction. In addition, the results show that OMG-conservative is more suitable than OMG-optimistic in this scenario since this is a general-sum game. The baselines based on action modeling, LIAM and Naïve OM, demonstrate comparable performance, whereas D3QN without opponent modeling, exhibits subpar results. In the predator-prey environment, the agent acts as the predator and collaborates with the other two uncontrolled predators to catch the prey. The results in Figure 4(b) show that OMG obviously outperforms action modeling methods, which demonstrates that OMG can also work efficiently in continuous state space. PPO without opponent modeling can hardly improve performance in training due to the non-stationarity caused by opponents. OMG-optimistic slightly performs better than OMG-conservative because OMG-optimistic is suitable for the cooperative game.

## 5.3 Generalization to Unknown Opponents

We evaluate the generalization of OMG in a more complex multi-agent environment, SMAC, which enables the opponents to exhibit more diverse policies. The experimental results of *8m*, *3s_vs_5z* and *2c_vs_64zg* are shown in Figure 5. Without opponent modeling, IQL struggles to adapt to various unknown opponents, resulting in poor performance, especially when the opponents are *non-homologue*. This underscores the effectiveness of opponent modeling in autonomous agent tasks. LIAM and Naïve OM, the action modeling methods, contribute to the team's improved win rate to some extent. The mediocre performance of OMG-conservative is attributed to its overly cautious subgoal selection. OMG-conservative is on par with IQL, which is consistent with the "conservative". OMG-optimistic surpasses the baseline methods, indicating that OMG-optimistic can generalize well to unknown collaborators through positive subgoal selection.

## 5.4 Ablation Study

The ablation study is conducted for the network structure of the inference model, subgoal selection, and hyperparameter horizon $H$. OMG uses CVAE as the inference model. Here, we instead employ supervised learning to train an inference model using the subgoal selector's output $\bar{g}_t$, obtained from either Equation (6) or Equation (7). This model is referred to as OMG-supervised. The results in the foraging environment are presented in Figure 6(a). The results indicate that OMG-optimistic and OMG-conservative outperform their counterparts, which is attributed to the enhanced adaptability of variational inference to the uncertainty in the opponent's policy. Dealing with multiple opponents employing distinct policies poses a challenge for supervised learning, as establishing a mapping relationship between historical trajectories and subgoals becomes intricate.

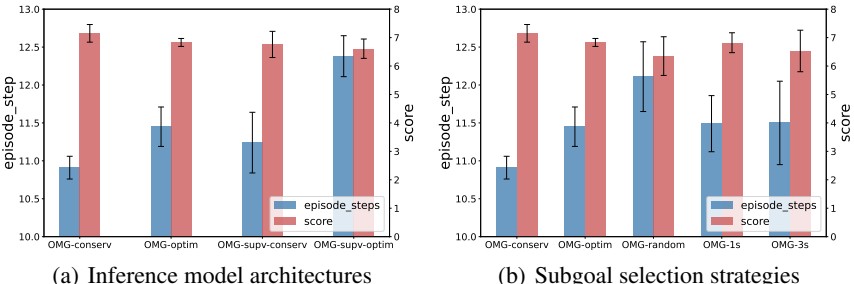

(a) Inference model architectures  (b) Subgoal selection strategies

Figure 6: Ablation study in Foraging. In (a), methods on the X-axis labeled with "supv" indicate that the inference model uses an MLP instead of a CVAE. In (b) OMG-random, OMG-1s, and OMG-3s represent subgoals selected from the opponent's future states: randomly, at the next step, and at the third step, respectively.

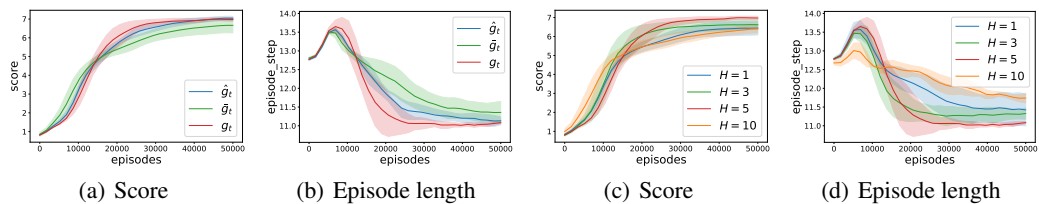

(a) Score  (b) Episode length  (c) Score  (d) Episode length

Figure 7: Ablation study of OMG in Foraging. (a) and (b) compares OMGs with different subgoal inputs for policy learning. (c) and (d) show ablation study for the hyperparameter horizon $H$.

In the OMG, the subgoal is selected by choosing the state within the future $H$ steps that either maximizes or minimizes the value function $V(s, g)$. To explore the impact of different subgoal selection strategies, we introduced three alternatives: random selection within the $H$ steps (OMG-random), selecting the first step as the subgoal (OMG-1s), and selecting the third step as the subgoal (OMG-3s). The results, presented in Figure 6(b), suggest that the choice of subgoal selection strategy significantly affects performance, with OMG's strategy leading to more effective training compared to the alternatives. We also observe that the subgoal often remains constant over consecutive time steps for OMG-supervised. Further details can be found in Appendix D.

We further investigate our design choice on the subgoal selection for the policy. During the policy update, Equation (8) (*i.e.*, $g$) is utilized. As $p_\psi$ is pre-trained and fixed during the update phase, $\bar{g}$ remains stable. On the other hand, $\hat{g}$, which represents the inferred subgoal when executing the policy, also stabilizes as the training steps increase. Thus, we choose a gradual transition of $g$ from $\bar{g}$ to $\hat{g}$, which should help avoid instability during the training of the subgoal inference model. Here we perform the experiments in the foraging environment with different subgoal inputs for the policy, *i.e.*, $g, \hat{g}, \bar{g}$. As shown in Figure 7(a) and Figure 7(b), OMG with $g$ indeed shows faster and better convergence.

The parameter $H$ denotes the horizon of the subgoal selector. The ablation experiment results are shown in Figure 7(c) and Figure 7(d). It is observed that an appropriate horizon value is neither excessively high nor excessively low. When $H = 1$, it is essentially equivalent to combining with QSS [9] and opponent modeling, which can be interpreted as another way of action modeling. However, if $H$ is set too large, such as $H = 10$, the agent may skip important states in the trajectory, leading to a degradation in performance. Therefore, selecting an appropriate value for $H$ is crucial in achieving satisfactory results.

## 5.5 Inferred Subgoal Analysis

We analyze the predictive performance of the opponent model. In Figure 8(a), we plot the ratio of that an opponent's future trajectory passes through the opponent's subgoal state, termed subgoal hit ratio. The subgoal state is reconstructed by the inferred subgoal $\hat{g}$ using the decoder of the subgoal prior model. The subgoal hit rate gradually improves during training, which indicates that

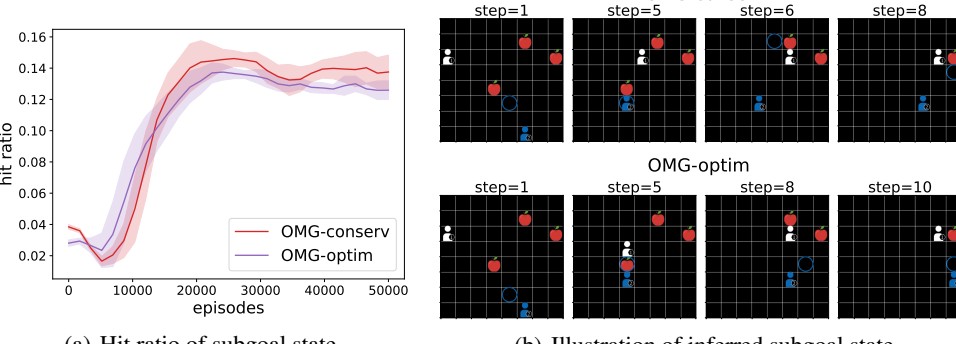

(a) Hit ratio of subgoal state  (b) Illustration of inferred subgoal state

Figure 8: Subgoal analysis of OMG in Foraging. The subgoal hit rates for OMG-conservative and OMG-optimistic are shown in Figure 8(a). In Figure 8(b), the agent controls player 1 (white), and the opponent controls player 2 (blue). The blue circle represents the state obtained through the reconstruction of the subgoal inferred by the agent. The figure illustrates the difference between OMG-conservative and OMG-optimistic under the same initial state and opponent policy.

the subgoal-based opponent modeling is able to predict the future state of the opponent. OMG tends to predict the opponent's future state several steps ahead as the subgoal, rather than focusing solely on the next step. This kind of prediction requires validation over multiple steps, and the agent policy conditioned on the predicted subgoal may also influence the behavior of the opponent. These make it challenging to verify the predicted subgoal. Consequently, the overall hit ratio remains at a moderate level at the end of training. There is a small gap between the subgoal hit rates of OMG-conservative and OMG-optimistic, which leads to a longer episode length for OMG-optimistic than OMG-conservative, as illustrated in Figure 8(b). The root cause lies in the differences in the subgoal selection between OMG-conservative and OMG-optimistic. More details can be found in Appendix E.

## 6 Conclusion and Limitation

In this paper, we introduce OMG, a novel method for opponent modeling based on subgoal inference. OMG is a simple and efficient opponent modeling method and can be combined with various RL algorithms. Unlike most opponent modeling methods, which primarily focus on predicting the opponent's actions, OMG focuses on modeling the opponent's subgoals. Specifically, it leverages the value function of the policy to guide the selection of subgoals, which yields two variants of OMG for cooperative and general-sum games, respectively. Empirical results demonstrate the remarkable performance achieved by OMG, as compared to baselines based on action modeling, and that OMG exhibits better generalization when cooperating with opponents with unknown policies. We analyze the subgoals obtained by the inference model, and the results show they closely correlate with the opponent's trajectory. The limitation of OMG is it cannot handle open multi-agent systems where agents may enter and leave during the interaction. This is left for future work.

### Acknowledgments

This work was supported by NSFC under Grant 62450001 and 62476008. The authors would like to thank the anonymous reviewers for their valuable comments and advice.

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

# A   Analysis of $(s, g)$

In opponent modeling, we usually build $(s, g)$ and $(s, a^{-i})$ by observing the opponent's action trajectories. We construct a tree to describe the trajectories of the opponent's action sequences, as shown in Figure 9. The non-leaf nodes and edges represent the state and opponent's action respectively. Without loss of generality, we simplify the problem by using a complete tree with the leaf node as goal. The length of the action sequences is $k$ and the opponent action space is denoted as $A$. We compare the number of $(s, a)$ and $(s, g)$ that can be observed via trajectories, and their sets are denoted as $\mathcal{S}_a$ and $\mathcal{S}_g$ respectively. The sizes of $\mathcal{S}_a$ and $\mathcal{S}_g$ as:

$$card(\mathcal{S}_a) = \sum_{l=0}^{k-1} \sum_{s \in \mathcal{S}^{(l)}} n_A = \frac{n_A^k - 1}{n_A - 1} n_A$$

$$card(\mathcal{S}_g) = \sum_{l=0}^{k-1} \sum_{s \in \mathcal{S}^{(l)}} \sum_{g \in G} \mathbb{I}(s \to g)$$

$$\leq |G| + n_A \cdot \frac{|G|}{n_A} + \cdots + n_A^{k-1} \cdot \frac{|G|}{n_A^{k-1}}$$

$$= k|G|,$$

where $\mathcal{S}^{(l)}$ represents the set of all states of depth $l$ in the tree. $s \to g$ means $g$ is reachable from $s$. $n_A$ is the size of $A$. Let $card(\mathcal{S}_g) \leq card(\mathcal{S}_a)$, we get a bound over $|G|$, as Equation (9). When the goal number of our method is within the bound, the number of expanded states can be significantly reduced, which means the RL algorithm learns faster than those methods based on action modeling.

$$card(\mathcal{S}_g) \leq card(\mathcal{S}_a) \Rightarrow |G| \leq \frac{n_A}{k} \frac{n_A^k - 1}{(n_A - 1)} = \frac{n_A}{k} |\mathcal{S}|. \tag{9}$$

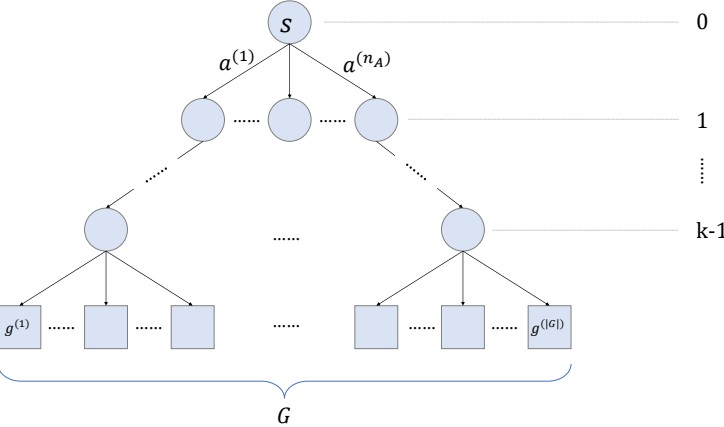

Figure 9: Illustration of opponent's decision tree. Circles, edges, and squares represent state nodes, action, and goal nodes respectively.

When $|G|$ is below $n_A/k$ times the number of observed states, the goal-based opponent modeling method proves more advantageous compared to the methods based on action modeling. Consequently, this criterion can be met by maintaining a relatively modest value for $k$. Due to our method favoring the adoption of extreme values as goal states, a limited quantity of such states exist. So, it is loosely bound of $|G|$ for OMG.

# B  Experiments settings

**Multi-Agent Environments.** Foraging environment [2, 4] is an $8 \times 8$ gridworld with full observation, containing two players: the agent and the opponent. At the beginning of each round, the players and three foods are randomly generated in the environment. The goal of the agent is to collect all foods as quickly as possible. The agent can move in four directions or pick up the food. The agent must judge the opponent's target food as soon as possible to avoid futile actions for the same food.

Predator-Prey [20] is a three-against-one multi-agent environment with a continuous space. Three predators coordinate to touch the prey, and all participants have full observation. The agent acts as one of the predators, and the opponents are the other two predators and the prey, which leads to the non-stationarity of the environment from the agent's view despite not belonging to one camp. The agent aims to maximize its reward and therefore needs to collaborate with the other two predators to complete the encirclement and cut the prey's escape route.

SMAC [35] is a high-dimensional partial observation complex environment for research in the field of collaborative MARL based on StarCraft II. The agent joins a set of opponents with unknown policies to accomplish the task. The only way to accomplish the task is to collaborate with the other agents. The agent's goal is to complete the task with a group of opponents controlled by unknown policies.

**Opponent.** The autonomous agent is trained in a multi-agent environment, where it interacts with the opponents controlled by a set of pre-trained policies. At the onset of each episode, the opponent's policy is selected randomly from the set. In the case of SMAC, the autonomous agent's index is also randomly determined. For Foraging, Predator-Prey, and SMAC environments, D3QN, PPO, and QMIX are used to train the opponents, respectively. All the opponents in the training set comprise 10 distinct policies.

In SMAC, the test set consists of 30 opponents with different policies, trained by the IQL, VDN [41], and QMIX [33]. In *8m*, the opponents are reorganized into three groups: *7 homologues*, *6 homologues*, and *7 non-homologues*. In *3s_vs_5z*, the opponents falls into two groups: *2 homologues* and *2 non-homologues*. Here, *homologue* refers to the policy from the same algorithm with the same parameters, and *non-homologue* represents the policy from two different algorithms. The remaining agent is controlled by OMG or baseline algorithms.

When assessing the performance of the autonomous agent in the SMAC with a test set, these opponents in the set are trained separately using IQL, VDN, and QMIX, with 10 instances for each training method. To illustrate the dissimilarity of the test opponent's policies, we utilize a set of identical states to acquire the action vectors of the policy in the test set. We visualize the action vectors, as demonstrated in Figure 10. The figure shows the diversity of test set policies employed by the test opponents. The test results are averaged over 100 episodes of fine-tuning, with 5 random seeds.

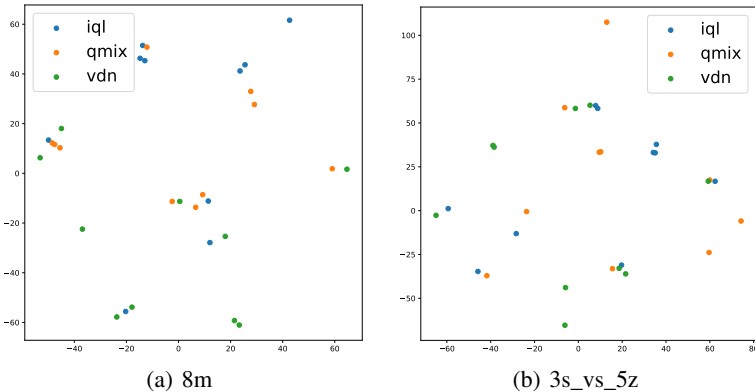

(a) 8m                    (b) 3s_vs_5z

Figure 10: The distribution of the opponent's policy for the test of generalization.

**Pre-train the subgoal's prior model.** The subgoal's prior model $p_\psi(\bar{g}|s^g)$ is a VAE that learns from a set of states that are collected while training opponents. The optimization objective of VAE is :

$$< \hat{\omega}, \hat{\psi} > = \underset{\omega, \psi}{\arg\max} \, \mathbb{E}_{g \sim q_\psi(g|s)} \Big[ \log p_\omega(s|g) \Big] - \text{KL}\Big( q_\psi(g|s) || \mathcal{N}(0, 1) \Big). \tag{10}$$

The decoder $p_\omega(s|g)$ is also used to reconstruct the subgoal state, as discussed in Section 5.5.

**Hyperparameters.** All hyperparameters are listed in Table 1.

Table 1: Hyperparameters

| | **Q-based RL** | Foraging(D3QN) | SMAC(IQL) | **Policy-based RL** | Predator-prey(PPO) |
|---|---|---|---|---|---|
| RL Algorithm | hidden units | MLP[64, 32] | RNN[64, 64] | hidden units | MLP[64, 32] |
| | activation function | ReLU | ReLU | activation function | ReLU |
| | optimizer | Adam | RMSProp | optimizer | Adam |
| | learning rate | 0.005 | 0.0005 | learning rate | 0.0005 |
| | target update interval | 100 | 200 | num. of updates | 10 |
| | epsilon start | 0.5 | 0 | value discount factor | 0.99 |
| | epsilon end | 0.95 | 0.95 | GAE parameter | 0.99 |
| | epsilon anneal time | 4500 | 50000 | clip parameter | 0.115 |
| | batch size | 32 | 32 | max grad norm | 0.5 |
| | buffer size | 5000 | 5000 | | |
| Opponent model | hidden units | MLP[64, 32] | MLP[64, 32] | | MLP[64, 32] |
| | learning rate | 0.001 | 0.001 | | 0.001 |
| | subgoal horizon $H$ | 5 | 10 | | 5 |
| | KL weight | 0.001 | 0.001 | | 0.001 |
| | $\Delta\eta$ | 0.001 | 0.001 | | 0.001 |
| | $\epsilon$ start | 0.5 | 0.5 | | 0.5 |
| | $\epsilon$ anneal time | 50000 | 50000 | | 50000 |

The computational resources for the experiments are as follows: the CPU is Intel(R) Xeon(R) Platinum 8280 CPU @ 2.70GHz, and the GPU is A100-PCIE-40GB.

## C  Performance of CTDE agent in autonomous agent task

The motivation of this paper is to address the autonomous agent through opponent modeling. The question is, can the CTDE agents be adapted to do tasks like autonomous agents? The two domain are fundamentally different, and the training method of CTDE doesn't work well in such situations because it hasn't been exposed to a variety of opponents during its training. We conducted tests in *8m* where QMIX acts as the agent with opponents of test set.

Table 2: Test performance on 8m

| opponent type | 7 non–homologue | 6 homologue | 7 homologue |
|---|---|---|---|
| QMIX | 21.6% | 65.6% | 61.0% |
| OMG-optim | 70.5% | 86.8% | 90.2% |

This training paradigm employed by QMIX leads to a lack of generalization for different opponents. Using the same training methodology for QMIX as OMG leads to a degradation in IQL,which already serves as one of the existing baselines. The test results indicate that opponents trained using different methods and seeds are not homogeneous, which poses challenges for cooperation.

# D  $s_g$ **selection frequency**

In Equation (6) and Equation (7), subgoals are selected from $\mathcal{N}_t^H = \{s_{t+k} | 1 \leq k \leq H\}$, where $\mathcal{N}_t^H$ is a sliding window. If the value is monotonic along the trajectory, a possible case is that different subgoals $\bar{g}$ are chosen at each step. We used 100 trajectories and counted the selection frequency within the trajectory, as shown in Table 3.

Table 3: The proportion of each $s_{t+k}$ in $\mathcal{N}_t^H$.

| $k$ | 1 | 2 | 3 | 4 | 5 |
|---|---|---|---|---|---|
| proportion | 35.4% | 20.0% | 22.8% | 19.3% | 2.5% |

The results suggest that in multi-agent settings, characterized by cooperative and competitive interactions among agents, the value function displays multiple peaks along the trajectory. Additionally, the selected subgoals exhibit a certain level of continuity over several steps.

# E   **Details for Inferred Subgoal Analysis**

**Subgoal hit ratio.** Define the opponent's trajectory sequence from $t = 0$ to $t = T$ as $\mathcal{T} = (s_0, s_1, \ldots, s_T)$, and the agent's prediction from time $t = 0$ to $t = T - 1$ as $\hat{\mathcal{T}} = (\hat{s}_0, \hat{s}_1, \ldots, \hat{s}_{T-1})$. Let $\mathcal{T}_i = (s_i, s_{i+1}, \ldots, s_T)$ represent the opponent's trajectory from step $i$ onward.

The subgoal hit ratio is calculated as follows:

$$\text{subgoal hit ratio} = \frac{|\{\hat{s}_t \mid \hat{s}_t \in \mathcal{T}_t, t = 0, 1, \ldots, T - 1\}|}{|\mathcal{T}|}$$

where $\{\hat{s}_t \mid \hat{s}_t \in \mathcal{T}_t, t = 0, 1, \ldots, T - 1\}$ represents the set of predicted states that are also present in the opponent's future trajectory starting from the current time step $t$.

For example, the opponent's trajectory is $\mathcal{T} = (s_1, s_2, s_3, s_4, s_5)$. For each time step, the relevant opponent trajectory is: $\mathcal{T}_0 = (s_1, s_2, s_3, s_4, s_5)$, $\mathcal{T}_1 = (s_2, s_3, s_4, s_5)$, $\mathcal{T}_2 = (s_3, s_4, s_5)$, $\mathcal{T}_3 = (s_4, s_5)$. The agent's predictions are $\hat{\mathcal{T}} = (s_3, s_1, s_5, s_5)$. The matched predicted states are $\{s_3, s_5\}$.

Thus, the hit ratio is calculated as:

$$\text{subgoal hit ratio} = \frac{|\{s_3, s_5\}|}{|\{s_1, s_2, s_3, s_4, s_5\}|} = \frac{2}{5} = 0.4$$

The complete trajectory of the example in Figure 8(b) is shown below:

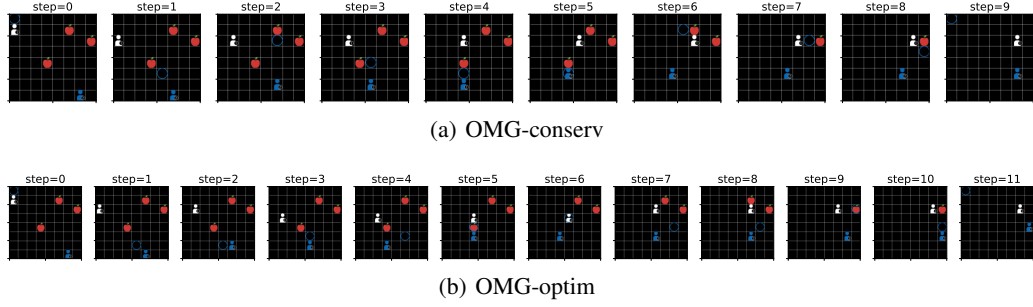

(a) OMG-conserv

(b) OMG-optim

Figure 11: Illustration of inferred subgoal state

# F  OMG based on PPO

---

**Algorithm 2** OMG based on PPO

---

1: ***Preparation:***
2: Interact with $\nu$ opponents to collect $s$ and train the prior model $f_\psi$
3: Initialize subgoal inference model parameters $\tau$ and $\theta$
4: Initialize policy parameters $\delta$ and value function parameters $\varphi$
5: **for** k=0,1,2,... **do**
6:     ***Interaction phase***
7:     Observe state $s$ and last opponent's action $a^{-i}$
8:     Infer the subgoal $\hat{g}$ by subgoal inference model $q_\phi(g|\tau)$
9:     Choose action $a$ by $\pi_{\delta_k}(\cdot|s,\hat{g})$
10:    Store experience $(s, a, a^{-i}, r)$ in buffer $\mathcal{D}_k$
11:    ***Update phase***
12:    Calculate prior subgoal $\bar{g}$ by Equation (6) or Equation (7)
13:    Calculate subgoal $g$ by Equation (8)
14:    Update policy parameters by

$$\delta_{k+1} = \arg\max_\delta \frac{1}{|\mathcal{D}_k|T} \sum_{\tau \in \mathcal{D}_k} \sum_{t=0}^{T} \min\big(\frac{\pi_\delta(a_t|s_t)}{\pi_{\delta_k}(a_t|s_t)} A^{\pi_{\delta_k}}(s_t, a_t), g(\epsilon, A^{\pi_{\delta_k}}(s_t, a_t))\big) \quad (11)$$

15:    Update value parameters by

$$\varphi_{k+1} = \arg\min_\varphi \frac{1}{|\mathcal{D}_k|T} \sum_{\tau \in \mathcal{D}_k} \sum_{t=0}^{T} (V_\varphi(s_t) - \hat{R}_t)^2 \quad (12)$$

16:    Update inference model $q_\phi$ and $p_\theta$ by Equation (5)
17: **end for**

---

