# OpenReview forum: "Opponent Modeling based on Subgoal Inference"
_NeurIPS.cc/2024/Conference — NeurIPS 2024 poster_

### Official Review · Reviewer_FFHk · 2024-06-20

**Soundness:** 3
**Presentation:** 3
**Contribution:** 3
**Rating:** 6
**Confidence:** 3

**Summary:**

The paper proposes an algorithm for learning policy in a multi-agent environment that predicts subgoal intents of other agents. In this way, the authors address the non-stationarity problem in multi-agent learning. Experiments in three environments show an advantage of the approach over baselines.

**Strengths:**

- The proposed method is simple and interesting.

- The paper is well-organized and mostly clear.

- Pseudocode and data flow charts are helpful.

**Weaknesses:**

- Experiments: It's a bit disappointing that in most of the presented experiments, the confidence intervals of OMG and baselines generally overlap (except in the predator-prey environment). I understand that controlling a single agent makes it hard to jump from 10% to 90%. However, I suggest choosing environments that amplify the impact of a single agent. For instance, the 3s_vs_5z environment seems well-chosen (hence the highest advantage). In either case, increasing the number of trials to make confidence intervals tighter would also help.

- The authors are sometimes overconfident in their claims. "Obviously, the complexity and uncertainty of predicting the action sequence are much higher than the goal itself" -- this is not always obvious. I agree that in some cases it is true, but there are also many domains where minor action changes lead to completely different states, resulting in much higher uncertainty in predicting subgoals. Instead of suggesting that OMG is always better, I recommend analyzing in which contexts OMG is better. Also, "... the results show they closely correlate with the opponent's trajectory" -- I acknowledge the reported 12-14% hit ratio, particularly given that the intents may change, but I'm not convinced that's enough evidence to claim "closely correlated".

- Equations 6-7: It's confusing that g in V(s_t, g) is the opponent's subgoal, while the notation resembles the standard goal-conditioned value function. I suggest replacing it with V(s_t, g^o), similar to a^o, or using any other indication that it's the opponent's subgoal. Additionally, to analyze the impact of choosing either (6) or (7), you should include an ablation (even a single experiment with just the results provided) in which you compare to naive choices, such as a random state or a fixed-distance state (e.g., s_t+3). This is especially important given that in Appendix D you report that 80% of subgoals are <=3 steps.

Minor comments:

- l.139: Technically, by modeling the opponents' subgoals, you don't gain any new information because everything is computed from the observation.

- l.148: Why is |G| finite? I think you didn't assume the finiteness of anything, including the action space.

- l.154-157: This claim needs stronger support to be that general. The experiment you show here is an illustrative example, too small to frame the takeaway so broadly.

- l.160: Why are there fewer subgoal tuples? I understand that both methods create exactly one datapoint for every step. If that's because of duplicates in the buffer, then this example may not be fully relevant (as in other environments, it is nearly impossible to repeat exactly the same state).

- l.297: I'd also mention here which value of H you've chosen.

**Questions:**

What is the speed of training and inference of OMG compared to Naive OM and standard RL baselines (wall time)?

**Limitations:**

One limitation is mentioned in Section 6. Actually, I don't see why OMG cannot handle open systems. However, I'd add other limitations, such as: modeling subgoals instead of actions is much harder if the state space is much more complex than the action space (e.g., in video games). Additionally, predicting future states may be prohibitively challenging if the policy strongly depends on the opponent's moves (e.g., chess).

There is no potential negative societal impact of this work.

---

> ### Author Rebuttal · Authors · 2024-08-07
>
> Thanks for your valuable comments. As follows, we address your concerns in detail.
>
> > Experiments: It's a bit disappointing that in most of the presented experiments, the confidence intervals of OMG and baselines generally overlap (except in the predator-prey environment). I understand that controlling a single agent makes it hard to jump from 10% to 90%. However, I suggest choosing environments that amplify the impact of a single agent. For instance, the 3s_vs_5z environment seems well-chosen (hence the highest advantage). In either case, increasing the number of trials to make confidence intervals tighter would also help.
>
> While OMG does not show a significant advantage in the experiments, it maintains a modest but consistent edge. This is reasonable, considering that opponent-predicting ability is not the sole factor influencing performance.
>
> We have increased the number of runs to 10 seeds for Predator-prey, and the result presents in the PDF file of "global" response.
>
> > Equations 6-7: It's confusing that g in V(s_t, g) is the opponent's subgoal, while the notation resembles the standard goal-conditioned value function. I suggest replacing it with V(s_t, g^o), similar to a^o, or using any other indication that it's the opponent's subgoal. Additionally, to analyze the impact of choosing either (6) or (7), you should include an ablation (even a single experiment with just the results provided) in which you compare to naive choices, such as a random state or a fixed-distance state (e.g., s_t+3). This is especially important given that in Appendix D you report that 80% of subgoals are <=3 steps.
>
> Following your suggestion, we have added ablation experiments. In these experiments, OMG-random, OMG-1s, and OMG-3s represent subgoals selected from the opponent's future states randomly, at the next step, and at three steps ahead, respectively. The results present in the PDF file of "global" response.
>
> > l.139: Technically, by modeling the opponents' subgoals, you don't gain any new information because everything is computed from the observation. l.148: Why is |G| finite? I think you didn't assume the finiteness of anything, including the action space.
>
> Thank you for your correction. Indeed, these statements are imprecise. We are not limited to discrete spaces; in continuous spaces, the state space is infinite. We will correct this.
>
> > l.154-157: This claim needs stronger support to be that general. The experiment you show here is an illustrative example, too small to frame the takeaway so broadly.
>
> We understand your concern regarding the strength of the statement about the extension of Q-values with subgoals. However, we believe that illustrative example prove the extension of Q-table works. Additionally, similar pattern has been used in the literature [1].
>
> [1] Ashley Edwards, Himanshu Sahni, Rosanne Liu, Jane Hung, Ankit Jain, Rui Wang, Adrien Ecoffet, Thomas Miconi, Charles Isbell, and Jason Yosinski. Estimating q (s, s’) with deep deterministic dynamics gradients. In International Conference on Machine Learning, pages 2825–2835. PMLR, 2020
>
> > l.160: Why are there fewer subgoal tuples? I understand that both methods create exactly one datapoint for every step. If that's because of duplicates in the buffer, then this example may not be fully relevant (as in other environments, it is nearly impossible to repeat exactly the same state).
>
> In this experiment, we adopt the Q-learning algorithm. These tuples represent the cells in the Q-table.
>
> In the example shown in Figure 3, $(\mathcal{A} = \mathcal{A}^o = \{ \text{none}, \uparrow, \downarrow, \leftarrow, \rightarrow \})$ and $(\mathcal{G} = \{D_1, D_2\})$. In the Q-table, the number of entries for $(s, g, a)$ is $(10 \cdot |S|)$, which is fewer than the number of entries for $(s, a^o, a)$.
>
> Due to exploration, it is practically impossible to update all cells in the Q-table. During my experiments, I observed that the number of cells used in the Q-table with $(s, a^o, a)$ was indeed fewer.
>
> > l.297: I'd also mention here which value of H you've chosen.
>
> The hyperparameters are presented in Appendix B.In both Foraging and Predator-Prey, the subgoal horizon $H$ is 5. In SMAC, $H$ is also 10.
>
> Appendix B lists the hyperparameters. The subgoal horizon $H$ is 5 for Foraging and Predator-Prey, and 10 for SMAC.
>
> > What is the speed of training and inference of OMG compared to Naive OM and standard RL baselines (wall time)?
>
> The training and test times are shown in the table below. OMG requires additional training time to compute subgoals from the buffer, as described in Eq. 6 and Eq.7.
>
> |   | OMG | Naive OM | D3QN |
> |:-----:|:-----:|:-----:|:-----:|
> | training time (minutes) | 848 | 588 | 273 |
> | test time (minutes) | 38 | 36 | 28 |

---

> > ### Comment · Reviewer_FFHk · 2024-08-10
> >
> > Thank you for the clarifications and additional experiments.
> >
> > > ... the confidence intervals of OMG and baselines generally overlap ...
> > > We have increased the number of runs to 10 seeds for Predator-prey ...
> >
> > Actually, the predator-prey results were fine, I was more concerned about the results in foraging (where the confidence intervals are by an order of magnitude larger than the difference between OMG, naive OM, and LIAM) and partially about 8m. Could you please clarify what are the takeaways from the experiments in Foraging? If the algorithms differ mostly in terms of the number of solution steps, I'm not sure whether this is uniformly reinforced across the methods, or just a chance (given similar scores).
> >
> > Since you provide some experiments in SMAC, I suggest trying more of them instead, particularly: 2m_vs_1z, corridor, 2s_vs_1sc, 2c_vs_64zg. In those tasks the impact of a single agent would be much more visible, and, usually, crucial for the success of the whole team. It would be exciting to see that two agents learn a cooperative tactic to win, and then OMG enters and it's able to adapt to that tactic. Especially in corridor, where precise coordination is crucial. Do you think it should be the case? I'm not sure if OMG would be able to follow a subtle tactic off-the-shelf (and hence it may be worse than the original team of agents, which is totally fine), but based on your motivation it should perform better than other variants of opponent modelling, right?

---

> > > ### Author Response · Authors · 2024-08-10
> > > **Response to Reviewer FFHk**
> > >
> > > Thanks for your valuable comments.
> > >
> > > > Actually, the predator-prey results were fine, I was more concerned about the results in foraging (where the confidence intervals are by an order of magnitude larger than the difference between OMG, naive OM, and LIAM) and partially about 8m. Could you please clarify what are the takeaways from the experiments in Foraging? If the algorithms differ mostly in terms of the number of solution steps, I'm not sure whether this is uniformly reinforced across the methods, or just a chance (given similar scores).
> > >
> > > The results of Foraging experiments are shown in the following table:
> > > |   | score | episode_step |
> > > |:-----:|:-----:|:-----:|
> > > | Naive OM  | 6.95 ± 0.33 | 11.61 ± 0.41 |
> > > | LIAM | 6.99 ± 0.27 | 11.34 ± 0.37 |
> > > | D3QN | 6.35 ± 0.2 | 12.57 ± 0.19 |
> > > | OMG-conserv | 7.15 ± 0.31 | 10.91 ± 0.15 |
> > > | OMG-optim	 | 6.83 ± 0.14 | 11.45 ± 0.26 |
> > >
> > > In fact, the confidence intervals are not orders of magnitude apart. The chart may look a bit more exaggerated.
> > >
> > > The ability to predict opponent's action/subgoal is not the only factor that determines the performance. In the Foraging, the suboptimal strategy adjusts its trajectory towards the correct target after 1-2 steps of hysteresis. The baseline algorithms may realize that the target is invalid until the opponent reaches the target point near it, our method predicts this earlier, allowing for a earlier switch to an alternative target. However, due to the random initial positions, not every episode presents a dominant opportunity. A statistical difference of 0.5 at episode steps indicates a substantial advantage in this context.
> > >
> > > > Since you provide some experiments in SMAC, I suggest trying more of them instead, particularly: 2m_vs_1z, corridor, 2s_vs_1sc, 2c_vs_64zg. In those tasks the impact of a single agent would be much more visible, and, usually, crucial for the success of the whole team. It would be exciting to see that two agents learn a cooperative tactic to win, and then OMG enters and it's able to adapt to that tactic. Especially in corridor, where precise coordination is crucial. Do you think it should be the case? I'm not sure if OMG would be able to follow a subtle tactic off-the-shelf (and hence it may be worse than the original team of agents, which is totally fine), but based on your motivation it should perform better than other variants of opponent modelling, right?
> > >
> > > Thank you for your suggestion! OMG's advantage over other opponent modeling algorithms that predict actions lies essentially in its ability to predict further steps. Admittedly, not all tasks require this level of prediction. We'll follow your advice by testing on more maps and will share the results here.

---

> > > > ### Comment · Reviewer_FFHk · 2024-08-13
> > > >
> > > > Thank you for your response. I acknowledge the clarifications by increasing my rating. I'd be happy to increase it even further if the experiments were extended as I suggested, with a proper discussion of the results.

---

> > > > > ### Author Response · Authors · 2024-08-13
> > > > > **Response to Reviewer FFHk**
> > > > >
> > > > > Thank you for your acknowledgement of the previous reply. Some experimental results are as follows:
> > > > >
> > > > > |   | OMG-optim | LIAM | Naive OM |
> > > > > |:-----:|:-----:|:-----:|:-----:|
> > > > > | corridor | 0% | 0% | 0% |
> > > > > | 2c_vs_64zg | 64.4% ± 8.1% | 48.4% ± 6.2% | 41.2% ± 15.2% |
> > > > >
> > > > > On the corridor map, both OMG and the baseline models had a 0% win rate because the teammate policies trained by QMIX failed to master this hard task.
> > > > >
> > > > > On the 2c_vs_64zg map, OMG perform better than the baseline algorithms and even over QMIX's 58% win rate. We noticed that OMG often led to the unit sticking closer to the teammate to avoid getting surrounded by enemies.
> > > > >
> > > > > Other experiments are still running and we will provide them as soon as possible. Thank you again for your acknowledgement.

---

> > > > > > ### Comment · Reviewer_FFHk · 2024-08-13
> > > > > >
> > > > > > Thank you for the additional results. As you can see, this kind of evaluation amplifies the difference between OMG and other methods, hence allowing for more meaningful conclusions. Have you also checked the performance of OMG-conservative in the 2c_vs_64zg task?

---

> > > > > > > ### Author Response · Authors · 2024-08-13
> > > > > > > **Response to reviewer FFHk**
> > > > > > >
> > > > > > > Thank you again for your valuable suggestions. Indeed, OMG achieves better results on this kind of map. We haven't run OMG-conservative yet. According to previous experiments, OMG-optimistic usually performs better than OMG-conservative on SMAC. We'll complete the experiment but will exceed the discussion deadline.

---

> > > > > > > ### Author Response · Authors · 2024-08-14
> > > > > > > **Response to reviewer FFHk**
> > > > > > >
> > > > > > > Thank you for your review.
> > > > > > >
> > > > > > > I'd like to add another experiment result: On the 2s_vs_1sc map, OMG, LIAM, and Naive OM all achieved 100% win rate.

---

> > > > > > > ### Author Response · Authors · 2024-08-14
> > > > > > > **Response to reviewer FFHk**
> > > > > > >
> > > > > > > We apologize for not being able to complete the experiments you suggested before the discussion ended. However, the current results show that OMG-optimistic plays a significant role in such small team cooperation tasks. We will finish the remaining tests and include them in the revision. We hope you might consider adjusting the score. Thank you again for your insightful suggestions.

---

### Official Review · Reviewer_FFHE · 2024-07-08

**Soundness:** 4
**Presentation:** 3
**Contribution:** 3
**Rating:** 7
**Confidence:** 3

**Summary:**

For cooperative games and general-sum games, this paper proposes opponent modeling by inferring an opponent’s subgoals, rather than inferring actions. They empirically verify that this leads to either similar or better scores over baselines in Foraging (discrete grid game), Predator-Pray (continuous game), and SMAC (high dimensional), as well as a reduction in required timesteps per episode to reach comparable scores to baseline.

**Strengths:**

- The paper is well organized, provides ample context and prior research, and solid empirical evidence to prove its method.
- The idea of predicting subgoals rather than actions is rather intuitive, and the authors are clear in its formalization.
- The method is novel.
- The paper is fair and thorough in its comparison to baselines. Environments are diverse (discrete, continuous, high dimensional), baseline+OMG share the same neural network architecture, tests are run across 5 seeds and plots provide mean+standard deviation.
- A dedicated ablation studies section clarifies the reasoning behind the choice of VAE, horizon, and subgoal selection.
- Code is provided in supplementary material and will be open sourced upon acceptance

**Weaknesses:**

- Improvements over scores in baselines are minor in Foraging. The main difference between the baselines appears to be in the reduction of the episode steps. It would be interesting to see if there is a reduction in the average time steps towards the end of training in Predator-Prey.
- Minor typos and clarifications requested in Questions.

**Questions:**

- L152 — missing word?  “will reach the same ” → “will reach the same state”
- L159-160 — confusing sentence: “...resulting from the tuple (s, a^o, a) is more than numerous than (s,g,a) in the Q-table”
- L190 — confusing preposition: “state as…” → “state of”?
- L204 and L206-207 — both sentences here claim seemingly contradictory statements: “...adopting an optimistic strategy akin to the minimax strategy, which applies to cooperative games” seems to contradict the following statement “...leading to a conservative strategy similar to the minimax strategy, which is commonly used for general-sum games”. Are these both similar to minimax?
- Section 4.2: Subgoal Selector. This section is a bit hard to follow the reasoning for Eq. 6 and Eq. 7 for OMG-optimistic and OMG-conservative, even after referring to 4.1 Can you clarify in this section why you provide these two distinct manners for subgoal selection?
- Section 5, Figure 5 — It is not immediately clear what the X-axis is representing. The numbers in front of “non-homologue” and “homologue” do not have context (they are only clear from reading the Appendix). Can you add a sentence to the description of the Figure? Can you expand on the sentence: “The X-axis represents the opponent’s policies, and “homologue” refers to the policy learned by the same algorithm, while “non-homologue” represents different ones”?
- L294 — possible typo: “interrupted” → “interpreted”
- Figure 7b — OMG-conserv shows step 1, 5, 6, and 8; OMG-optim shows steps 1, 5, 8, 10. Is there a reason the steps are not matched? (i.e both showing 1,5,6,8 or 1,5,8,10)
- L299-300 — the wording of the definition for hit ratio is a bit unclear. Is this essentially the ratio of predicted opponent trajectory to actual opponent trajectory?
- Section 5.5: How do you determine the hit ratio if the subgoal is several steps ahead of the current state? For example, if a predicted subgoal is 3 steps north, the opponent could reach the subgoal in 3 steps or more steps. How many steps can an opponent take between the subgoal prediction state in order for it to still count as a hit?
- L479 — Typo: “ovservation” → “observation”
- L504 — Typo: “remained” → “remaining”

**Limitations:**

Yes.

---

> ### Author Rebuttal · Authors · 2024-08-07
>
> Thank you for acknowledging our novel contributions as well as raising valuable questions.
>
> > Improvements over scores in baselines are minor in Foraging. The main difference between the baselines appears to be in the reduction of the episode steps. It would be interesting to see if there is a reduction in the average time steps towards the end of training in Predator-Prey.
>
> The Predator-Prey has a fixed episode length. The score represents the number of times the predators touch the prey. Therefore, this result already reflects a reduction in the average touch time.
>
> > L152 — missing word? “will reach the same ” → “will reach the same state”
>
> Thank you for your correction. In this paper, goal $g$ represents a feature embedding of a future state. Therefore it is more accurate to use "will reach the same feature embedding" and we will fix it.
>
> > L159-160 — confusing sentence: “...resulting from the tuple (s, a^o, a) is more than numerous than (s,g,a) in the Q-table”
>
> In the example shown in Figure 3, $(\mathcal{A} = \mathcal{A}^o = \{ \text{none}, \uparrow, \downarrow, \leftarrow, \rightarrow \})$ and $(\mathcal{G} = \{D_1, D_2\})$. In the Q-table, the number of cells for $(s, g, a)$ is $(10 \cdot |S|)$, which is fewer than the number of cells for $(s, a^o, a)$.
>
> > L190 — confusing preposition: “state as…” → “state of”?
>
> This sentence implies that the selected state is being treated or used as a subgoal. Emphasizes the role or function of the state in the context of the model. We modify it in the revision to disambiguate.
>
> > L204 and L206-207 — both sentences here claim seemingly contradictory statements: “...adopting an optimistic strategy akin to the minimax strategy, which applies to cooperative games” seems to contradict the following statement “...leading to a conservative strategy similar to the minimax strategy, which is commonly used for general-sum games”. Are these both similar to minimax?
>
> In Line 204, the maximax strategy is used, while Line 206 employs the minimax strategy. These two strategies are different.
>
> > Section 4.2: Subgoal Selector. This section is a bit hard to follow the reasoning for Eq. 6 and Eq. 7 for OMG-optimistic and OMG-conservative, even after referring to 4.1 Can you clarify in this section why you provide these two distinct manners for subgoal selection?
>
> The maximax strategy aims to maximize the maximum possible gain. It is an optimistic approach that focuses on the best possible outcome. The formula is as follows:
>
> $a = \arg\max_{a_i \in \mathcal{A}} \left(\max_{a_o \in A_o} (P(a_i, a_o)) \right)$.
>
> The minimax strategy focuses on maximize the minimum gain. The formula is as follows:
>
> $a = \arg\max_{a_i \in \mathcal{A}} \left(\min_{a_o \in A_o} (P(a_i, a_o)) \right)$
>
> In Eq.6 corresponds to the inner "max" in the maximax strategy, and Eq.7 corresponds to the inner "min" in the minimax strategy. RL corresponds to the outer "max" of both.
>
> > Figure 7b — OMG-conserv shows step 1, 5, 6, and 8; OMG-optim shows steps 1, 5, 8, 10. Is there a reason the steps are not matched? (i.e both showing 1,5,6,8 or 1,5,8,10)
>
> The trajectories formed by the two algorithms have different lengths and cannot be matched exactly. We will release the full trace instead of the key steps in the appendix.
>
> > L299-300 — the wording of the definition for hit ratio is a bit unclear. Is this essentially the ratio of predicted opponent trajectory to actual opponent trajectory?
>
> > Section 5.5: How do you determine the hit ratio if the subgoal is several steps ahead of the current state? For example, if a predicted subgoal is 3 steps north, the opponent could reach the subgoal in 3 steps or more steps. How many steps can an opponent take between the subgoal prediction state in order for it to still count as a hit?
>
> For example, the opponent’s trajectory sequence from $ t=0 $ to $ t=4 $ is $(s_1, s_2, s_3, s_4, s_5)$. The agent's prediction from $ t=0 $ to $ t=3 $ is $(s_3, s_1, s_5, s_5)$. The hit ratio is calculated as $\frac{|\{s_3, s_5\}|}{|\{s_1, s_2, s_3, s_4, s_5\}|} = 0.4$. The predicted $s_1$ at $ t=1 $ is not counted because $s_1$ is not present in the trajectory sequence from $t>=1$. We'll add a detailed explanation in the appendix.
>
> > Section 5, Figure 5 — It is not immediately clear what the X-axis is representing. The numbers in front of “non-homologue” and “homologue” do not have context (they are only clear from reading the Appendix). Can you add a sentence to the description of the Figure? Can you expand on the sentence: “The X-axis represents the opponent’s policies, and “homologue” refers to the policy learned by the same algorithm, while “non-homologue” represents different ones”?
>
> > L294 — possible typo: “interrupted” → “interpreted”
>
> > L479 — Typo: “ovservation” → “observation”
>
> > L504 — Typo: “remained” → “remaining”
>
> Thank you for your suggestions and corrections. We will fix it in the revision.

---

> > ### Comment · Reviewer_FFHE · 2024-08-07
> >
> > Thank you for your response to my comments. I have read your rebuttal and will provide further comments soon, as needed.

---

> > > ### Comment · Reviewer_FFHE · 2024-08-11
> > >
> > > You have addressed my concerns and I have no further comments. I would like to maintain my score.

---

> > > > ### Author Response · Authors · 2024-08-12
> > > > **Response to reviewer FFHE**
> > > >
> > > > Thank you sincerely for your comments and appreciation.

---

### Official Review · Reviewer_fQcX · 2024-07-12

**Soundness:** 3
**Presentation:** 3
**Contribution:** 2
**Rating:** 5
**Confidence:** 4

**Summary:**

The paper is positioned within the ad-hoc teamwork problem in multi-agent systems, where an agent faces partners it has not seen before in training, and must learn to cooperate with them towards performing a task that benefits from cooperation. Authors deal with a specific requirement of ad-hoc teamwork that is opponent modelling: an agent's ability to model other agents in the system. They propose to model other agents at the level of their subgoals, inferred from the observed trajectories of interaction. They propose to learn a Q-function that also takes the inferred opponent sub-goal as input, thus making the policy of the protagonist agent conditioned on the sub-goal of the opponent. The empirical results seem to suggest minor improvements over previous opponent modelling works.

**Strengths:**

**Originality**
- I do not believe there is much strength in terms of originality.

**Quality**
- The submission seems technically sound to me _except_ for the definition of a sub-goal. A sub-goal is defined as a feature embedding of a future state. This is not sound to me (see Weaknesses).

**Clarity**
- The clarity of the submission is slightly above average.

**General Comments**
As someone from this particular niche, I believe the paper is studying an important problem. Overall, I believe their approach of augmenting state-spaces with inferred information about the opponents is perhaps an under-explored direction.

**Weaknesses:**

**Originality**
- I do not believe that goal and sub-goal inference for modelling agents is original. The paper is missing an entire line of research in their related works here, namely the inverse planning literature. This community has been studying inferring goals and sub-goals from trajectories through probabilistic modelling of other agents for a long time now (e.g. see [1,2] as a starting point and follow the thread of references and citations).  In addition, inferring and modelling goals of others is not original also in the ad-hoc teamwork literature (see [3,4]). Others also considered goal recognition between agents actively [5]. These are only some of the papers in this line of work, as there are many more. I am surprised to see none of these works are cited. I would like to see a discussion on what is the novelty/originality left after this literature is accounted for. Conditioning the Q-function on the goal of other is also not an original idea, but this is not the main claim any way.

Additionally, the problem of learning to cooperate with previously unseen teammates has been defined and named as "ad-hoc teamwork" in 2010 [6]. Since it is literally the problem setting of the authors, I am surprised there is no mention of this and the original paper is not cited.

[1] Zhi-Xuan T, Mann J, Silver T, Tenenbaum J, Mansinghka V. Online bayesian goal inference for boundedly rational planning agents. Advances in neural information processing systems. 2020;33:19238-50

[2] Ying L, Zhi-Xuan T, Mansinghka V, Tenenbaum JB. Inferring the goals of communicating agents from actions and instructions. InProceedings of the AAAI Symposium Series 2023 (Vol. 2, No. 1, pp. 26-33).

[3] Melo FS, Sardinha A. Ad hoc teamwork by learning teammates’ task. Autonomous Agents and Multi-Agent Systems. 2016 Mar;30:175-219.

[4] Chen S, Andrejczuk E, Irissappane AA, Zhang J. ATSIS: achieving the ad hoc teamwork by sub-task inference and selection. InProceedings of the 28th International Joint Conference on Artificial Intelligence 2019 Aug 10 (pp. 172-179).

[5] Shvo M, McIlraith SA. Active goal recognition. InProceedings of the AAAI Conference on Artificial Intelligence 2020 Apr 3 (Vol. 34, No. 06, pp. 9957-9966).

[6] Stone P, Kaminka G, Kraus S, Rosenschein J. Ad hoc autonomous agent teams: Collaboration without pre-coordination. InProceedings of the AAAI Conference on Artificial Intelligence 2010 Jul 5 (Vol. 24, No. 1, pp. 1504-1509).

**Quality**
- The experimental section only compares to two relatively simple opponent modelling works LIAM and "Naive OM". However, the authors for instance literally cite the "Machine Theory of Mind" paper and the "Modelling others using oneself in multi-agent reinforcement learning" in related works. Surely these are the closest competitors to their method. Additionally, some of the papers listed above under originality are also close competitors. I believe you need more than two methods in opponent modelling baselines. Especially the two methods compared are by no means considered the state-of-the-art.

**Questions:**

Q1: "_However, focusing on the opponent’s action
is shortsighted, which also constrains the adaptability to unknown opponents in complex tasks_"

What do you mean by "shortsighted"? How does that _constrain_ the adaptability? These are strong statements but are not concretised at all.

Q2: "_...bridge the information gap between agents_"

This concept of _information gap_ appeared out of nowhere. It is not explained either. What does this even mean? It needs to be clarified.

Q3: "_Autonomous agents, different from those jointly trained, can act autonomously in complex and dynamic environments..._"

You seem to be conflating autonomy with decentralized training or being self-interested here. The definition of autonomy does not preclude centralised training. I can have a set of autonomous agents train with privileged information, yet deploy them to act autonomously.

Q4: "_Although a lot of the existing methods concentrate on modeling the opponent’s actions, we argue that such an approach is short-sighted, pedantical, and highly complex._"

Again, lots of sweeping and strong statements with no backing or clarification. What does short-sighted, pedantical, and highly complex mean? This does not sound professional or scientific. If you are going to actually complain about an entire line of literature, you need to make your statement more concrete.

Q5: "_Generally, modeling an opponent’s actions just predicts what it will do at the next step. Intuitively, it is more beneficial for the agent to make decisions if it knows the situation of the opponent several steps ahead_"

This statement seems to ignore the trajectory prediction literature entirely, and tries to get away with it by saying "generally".

Q6: "_Other methods that claim to predict the opponent’s goal [28 , 29], but without explicitly making a connection to the opponent’s goal..._"

What does this even mean? Also, see the inverse planning / goal recognition works cited in Weaknesses and their references/citations for works that predict goals explicitly.

Q7: "_Unlike these methods, in this paper, we consider the most common..._"

I would argue that the setting where other _autonomous_ agents are also learning and adapting to our agent is the more common setting, and the fixed opponent policy is a big simplification of it. This can be justified in certain cases, but it certainly is not the _most common,_ except that it is more common in literature because it is simpler. But if the harder and more realistic problem is already being tackled, I do not see why this point is a plus.

Q8: The $\pi^o, a^o$ notation is confusing. I would recommend using the pre-established notational norms in game theory / MARL, $\pi^{-i} , a^{-i}$ for all agents except $i$.

Q9: "_Opponent modeling typically predicts the actions of other agents to address the non-stationary problem_."

At this point, I am a little confused. You have said "_Unlike these methods, in this paper, we consider the most common setting where opponents have unseen, diverse, but **fixed policies** during test_." So the opponents have fixed policies during test. Then there is absolutely no non-stationarity during test time. In fact, since $\pi^o$ is fixed within a test episode, the problem the agent is facing in test time is in fact an MDP. So you are only dealing with the setting where at each episode the agent might be facing a new MDP. How is this different from online single-agent reinforcement learning then?

**Limitations:**

Yes.

---

> ### Author Rebuttal · Authors · 2024-08-07
>
> ## Part Ⅰ
>
> Thank you for valuable comments. Below is a detailed response to your question, addressing each point individually.
>
> > I do not believe that goal and sub-goal inference for modelling agents is original. The paper is missing an entire line of research in their related works here, namely the inverse planning literature. This community has been studying inferring goals and sub-goals from trajectories through probabilistic modelling of other agents for a long time now (e.g. see [1,2] as a starting point and follow the thread of references and citations). In addition, inferring and modelling goals of others is not original also in the ad-hoc teamwork literature (see [3,4]). Others also considered goal recognition between agents actively [5]. These are only some of the papers in this line of work, as there are many more. I am surprised to see none of these works are cited. I would like to see a discussion on what is the novelty/originality left after this literature is accounted for. Conditioning the Q-function on the goal of other is also not an original idea, but this is not the main claim any way. Additionally, the problem of learning to cooperate with previously unseen teammates has been defined and named as "ad-hoc teamwork" in 2010 [6]. Since it is literally the problem setting of the authors, I am surprised there is no mention of this and the original paper is not cited.
>
> We will add more relevant references to enhance the related works section. Below, we explain the differences between our work and the references you mentioned to highlight our contributions and originality.
>
> Our method represents an innovation in the field of opponent modeling. As described, OMG is applicable to both cooperative games and general-sum games, and is not limited to "ad-hoc teamwork" [3,4,6]. OMG emphasizes methodological innovation rather than focusing solely on the ability to solve specific tasks.
>
> Additionally, inferring goals is a straightforward concept and is commonly addressed in numerous papers [5, 7, 8, 9]. However, in the context of opponent modeling for autonomous agents, this method is novel. Unlike the method in [2], which requires agents to communicate with each other, our setting involves autonomous agents that cannot communicate with other agents. Model-based methods [1] require constructing an environmental model and involve extensive computation for planning during execution. This approach differs from the model-free method adopted in this work.
>
> > The experimental section only compares to two relatively simple opponent modelling works LIAM and "Naive OM". However, the authors for instance literally cite the "Machine Theory of Mind" paper and the "Modelling others using oneself in multi-agent reinforcement learning" in related works. Surely these are the closest competitors to their method. Additionally, some of the papers listed above under originality are also close competitors. I believe you need more than two methods in opponent modelling baselines. Especially the two methods compared are by no means considered the state-of-the-art.
>
> OMG differs from traditional action prediction methods in opponent modeling by introducing a new approach based on subgoal inference. Therefore, we selected related methods as baselines. To the best of our knowledge, we have compared state-of-the-art (SOTA) methods in this domain. Below, we will explain why certain methods were not included as baselines.
>
> [1] describes a planning algorithm using "Sequential Inverse Plan Search," which is fundamentally different from the model-free algorithm used in this paper. The only similarity is the concept of "goal inference." In [2], agents are able to communicate with each other, unlike our setting with autonomous agents that cannot communicate. References [3, 4, 6] focus on ad-hoc teamwork. The method in [3] does not use deep learning and is difficult to apply to complex environments like SMAC. [4] uses Goal-Conditioned RL (GCRL) and requires goals to be manually defined, whereas OMG does not. Additionally, goal-based reward shaping limits [4] to cooperative tasks. The problem addressed in [5] is active goal recognition, which is quite different from our approach of optimizing policies through opponent modeling. We have conducted a comprehensive survey of related work and compared all relevant baselines under fair conditions.
>
>
> [1] Zhi-Xuan T, Mann J, Silver T, Tenenbaum J, Mansinghka V. Online bayesian goal inference for boundedly rational planning agents. Advances in neural information processing systems. 2020;33:19238-50
>
> [2] Ying L, Zhi-Xuan T, Mansinghka V, Tenenbaum JB. Inferring the goals of communicating agents from actions and instructions. InProceedings of the AAAI Symposium Series 2023 (Vol. 2, No. 1, pp. 26-33).
>
> [3] Melo FS, Sardinha A. Ad hoc teamwork by learning teammates’ task. Autonomous Agents and Multi-Agent Systems. 2016 Mar;30:175-219.
>
> [4] Chen S, Andrejczuk E, Irissappane AA, Zhang J. ATSIS: achieving the ad hoc teamwork by sub-task inference and selection. InProceedings of the 28th International Joint Conference on Artificial Intelligence 2019 Aug 10 (pp. 172-179).
>
> [5] Shvo M, McIlraith SA. Active goal recognition. In Proceedings of the AAAI Conference on Artificial Intelligence 2020 Apr 3 (Vol. 34, No. 06, pp. 9957-9966).
>
> [6] Stone P, Kaminka G, Kraus S, Rosenschein J. Ad hoc autonomous agent teams: Collaboration without pre-coordination. AAAI 2010.
>
> [7] Silviu Pitis, Harris Chan, Stephen Zhao, Bradly Stadie, and Jimmy Ba. Maximum entropy gain exploration for long horizon multi-goal reinforcement learning. ICML, 2020.
>
> [8] Suraj Nair, Silvio Savarese, and Chelsea Finn. Goal-aware prediction: Learning to model what matters. ICML, 2020.
>
> [9] Menghui Zhu, Minghuan Liu, Jian Shen, Zhicheng Zhang, Sheng Chen, Weinan Zhang, Deheng Ye, Yong Yu, Qiang Fu, and Wei Yang. Mapgo: Modelassisted policy optimization for goal-oriented tasks. IJCAI, 2021.

---

> ### Author Response · Authors · 2024-08-07
> **Rebuttal(Part Ⅱ)**
>
> ## Part Ⅱ
>
> > Q1: "However, focusing on the opponent’s action is shortsighted, which also constrains the adaptability to unknown opponents in complex tasks". What do you mean by "shortsighted"? How does that constrain the adaptability? These are strong statements but are not concretised at all.
>
> Generally, modeling an opponent’s actions just predicts what it will do at the next step. A lot of opponent modeling papers[1,2,3] predict the next step action. Intuitively, it is more beneficial for the agent to make decisions if it knows the situation of the opponent several steps ahead. Predicting future states of the opponent have an advantage over predicting future actions. Just like our example in Sec 1 Para 3:
>
> *For example, to reach the goal point of $(2, 2)$, an opponent moves from $(0, 0)$ following the action sequence $<\uparrow,\uparrow,\rightarrow,\rightarrow>$ by four steps (Cartesian coordinates). There are also 5 other action sequences, \textit{i.e.,} $<\uparrow,\rightarrow,\uparrow,\rightarrow>, <\uparrow,\rightarrow,\rightarrow,\uparrow>, <\rightarrow,\uparrow,\uparrow,\rightarrow>, <\rightarrow,\uparrow,\rightarrow,\uparrow>, <\rightarrow,\rightarrow,\uparrow,\uparrow>$, that can lead to the same goal. Obviously, the complexity of the action sequence is much higher than the goal itself.*
>
> Similar conclusions are also verified by the experiments in Figure 3, and the corresponding analysis is given in Appendix A.1.
>
> [1] Georgios Papoudakis, Filippos Christianos, and Stefano Albrecht. Agent modelling under partial observability for deep reinforcement learning. Advances in Neural Information Processing Systems, 34:19210–19222, 2021.
>
> [2] Georgios Papoudakis and Stefano V Albrecht. Variational Autoencoders for Opponent Modeling in Multi-Agent Systems. arXiv preprint arXiv:2001.10829, 2020.
>
> [3] Haobo Fu, Ye Tian, Hongxiang Yu, Weiming Liu, Shuang Wu, Jiechao Xiong, Ying Wen, Kai Li, Junliang Xing, Qiang Fu, et al. Greedy when sure and conservative when uncertain about the opponents. In International Conference on Machine Learning, pages 6829–6848. PMLR, 2022.
>
> > Q2: "...bridge the information gap between agents". This concept of information gap appeared out of nowhere. It is not explained either. What does this even mean? It needs to be clarified.
>
> In Multi-Agent Reinforcement Learning (MARL), "bridging the information gap between agents" refers to enhancing communication and information sharing among agents. This process involves reducing the uncertainty or lack of information each agent has about the environment, as well as the states, actions, or intentions of other agents. By effectively bridging this gap, agents can make more informed decisions, coordinate better, and ultimately achieve more optimal outcomes in their collective tasks. The following literature [1] explores the importance and impact of information sharing among agents.
>
> [1] Tan, M. . "Multi-Agent Reinforcement Learning : Independent vs. Cooperative Agents." Proc. of 10th ICML (1993).
>
> > Q3: "Autonomous agents, different from those jointly trained, can act autonomously in complex and dynamic environments...". You seem to be conflating autonomy with decentralized training or being self-interested here. The definition of autonomy does not preclude centralised training. I can have a set of autonomous agents train with privileged information, yet deploy them to act autonomously.
>
> We will reorganize the discussion on "Autonomous agents and jointly trained" to eliminate any ambiguity. The aim here is to distinguish our method from those that are "jointly trained," as such methods struggle to generalize against opponents with varying policies, as results in Appendix C.
>
> > Q4: "Although a lot of the existing methods concentrate on modeling the opponent’s actions, we argue that such an approach is short-sighted, pedantical, and highly complex.". Again, lots of sweeping and strong statements with no backing or clarification. What does short-sighted, pedantical, and highly complex mean? This does not sound professional or scientific. If you are going to actually complain about an entire line of literature, you need to make your statement more concrete.
>
> These words "short-sighted, pedantical, and highly complex" are used to describe the weakness of "predicting action" in opponent modeling, which is also the advantage of our proposed "predicting subgoals". For example, A and B play the prophecy game, A can predict the next step of other agents, and B can predict the next 5 steps. Obviously, A is short-sighted compared with B.
>
> We give intuitive examples that illustrate the advantages of predicting subgoals. That example is detailed in Sec 1 Para 3. Similar conclusions are also verified by the experiments in Figure 3, and the corresponding analysis is given in Appendix A.1.

---

> ### Author Response · Authors · 2024-08-07
> **Rebuttal (Part Ⅲ)**
>
> ## Part Ⅲ
>
> > Q5: "Generally, modeling an opponent’s actions just predicts what it will do at the next step. Intuitively, it is more beneficial for the agent to make decisions if it knows the situation of the opponent several steps ahead". This statement seems to ignore the trajectory prediction literature entirely, and tries to get away with it by saying "generally".
>
> To the best of our knowledge, current literature on plan recognition [1, 2, 3], including the trajectory prediction methods you mentioned, requires substantial prior knowledge, such as hierarchical plan libraries or domain models. These methods, which depend on prior knowledge, are outside the scope of our discussion.
>
> [1] Blaylock, N., Allen, J., 2006. Fast hierarchical goal schema recognition. In: Proceedings of the 21st AAAI National Conference on Artificial Intelligence. pp. 796–801.
>
> [2] Sohrabi, S., Riabov, A., Udrea, O., 2016. Plan recognition as planning revisited. In: Proceedings of the 25th International Joint Conference on Artificial Intelligence. pp. 3258–3264.
>
> [3] Vered, M., Kaminka, G., 2017. Heuristic online goal recognition in continuous domains. In: Proceedings of the 26th International Joint Conference on Artificial Intelligence. pp. 4447–4454.
>
> [4] Tian, X., Zhuo, H., Kambhampati, S., 2016. Discovering underlying plans based on distributed representations of actions. In: Proceedings of the 15th International Conference on Autonomous Agents and Multiagent Systems. pp. 1135–1143.
>
> > Q6: "Other methods that claim to predict the opponent’s goal [28 , 29], but without explicitly making a connection to the opponent’s goal...". What does this even mean? Also, see the inverse planning / goal recognition works cited in Weaknesses and their references/citations for works that predict goals explicitly.
>
> In the domain of inverse planning, goals are indeed explicitly predicted. We want to emphasize that this paper discusses methods for predicting an opponent's goals within the context of opponent modeling. While both inverse planning and opponent modeling involve understanding other agents' behaviors, inverse planning focuses on inferring goals and intentions, whereas opponent modeling is concerned with predicting opponent behavior and gaining an advantage in the scenario.
>
> > Q7: "Unlike these methods, in this paper, we consider the most common...". I would argue that the setting where other autonomous agents are also learning and adapting to our agent is the more common setting, and the fixed opponent policy is a big simplification of it. This can be justified in certain cases, but it certainly is not the most common, except that it is more common in literature because it is simpler. But if the harder and more realistic problem is already being tackled, I do not see why this point is a plus.
>
> Thank you for your suggestion. We acknowledge that the term "most common" might be ambiguous, and we will revise it to avoid confusion. In reality, due to the high costs of continuous learning, it is rare for deployed agents to continually learning and adapting. Instead, periodic updates are more common. We believe that the setting where opponents have unseen, diverse, but fixed policies during testing is more prevalent at present.
>
> > Q8: The $\pi^o, a^o$ notation is confusing. I would recommend using the pre-established notational norms in game theory / MARL, $\pi^{-i}, a^{-i}$ for all agents except $i$.
>
> Thank you for your suggestion. We will revise it in the revision.
>
> > Q9: "Opponent modeling typically predicts the actions of other agents to address the non-stationary problem." At this point, I am a little confused. You have said "Unlike these methods, in this paper, we consider the most common setting where opponent s have unseen, diverse, but fixed policies during test." So the opponents have fixed policies during test. Then there is absolutely no non-stationarity during test time. In fact, since is fixed within a test episode, the problem the agent is facing in test time is in fact an MDP. So you are only dealing with the setting where at each episode the agent might be facing a new MDP. How is this different from online single-agent reinforcement learning then?
>
> Firstly, the sources of non-stationarity differ. In single-agent RL, non-stationarity arises from changes in the environment. In MARL, an additional challenge is the uncertainty from opponents' policies. These different sources lead to varying degrees and forms of non-stationarity, which should not be conflated. Secondly, online reinforcement learning is a broader concept. In our setting, the agent must quickly adapt to opponents' policy changes, which aligns with the goal of online RL to enable agents to adapt rapidly.

---

> > ### Comment · Reviewer_fQcX · 2024-08-10
> >
> > The response only partially address some of reservations. I choose to increase my score to acknowledge this, however I still do not think the paper is ready for publication and _at least_ must have a strong writing revision.

---

> > > ### Author Response · Authors · 2024-08-12
> > > **Response to reviewer fQcX (Part Ⅰ)**
> > >
> > > Thank you for your acknowledgement of the previous reply. Openreview is currently unable to continue uploading the revision version. We have included the major revisions in this reply.
> > >
> > > Sec 1 Para 1-2:
> > >
> > > Autonomous agents are systems capable of making decisions and acting independently in their environment, often operating without direct human intervention [3]. These agents can either cooperate with or compete against each other, depending on the context. In cooperative scenarios, many multi-agent reinforcement learning (MARL) methods [18,36,30,34] aim to bridge the information gap between agents [49] by training agents in a centralized manner, called centralized training with decentralized execution, enabling agents to work together seamlessly to accomplish cooperative tasks. Alternatively, fully decentralized methods[15,35] seek to break free from the constraints of centralized training, allowing agents to reach collaboration in a simpler and decentralized manner. In competitive scenarios, NFSP[13], PSRO[17], and DeepNash[26] employ self-play to train agents for equilibrium strategies, allowing agents to adapt and improve their policy. By considering how the agent affects the expected learning progress of other agents, LOLA[9] and COLA[44] apply opponent shaping to this setting. Overall, these methods focus on training agents in a way that accounts for their interactions, resulting in a set of policies that enable effective collaboration or competition within a group of agents.
> > >
> > > While the above methods emphasizes the collective behavior of agents, it is also crucial to consider the role of individual agents, particularly self-interested agents, in these multi-agent environments. A self-interested agent[50,51] operates with the primary goal of maximizing its own benefits, even when interacting with other agents. When the objectives of a self-interested agent align with those of the team, this scenario falls under ad-hoc teamwork[52,53,54]; however, in more general cases, these interactions are framed as noncooperative games [56,55]. A key technique for self-interested agents in such settings is *opponent modeling*[3,57], which enables them to analyze and predict the actions, goals, and beliefs of other agents. By modeling the intentions and policies of other agents, the training process of the agent might be stabilized [24]. Many studies rely on predicting the actions [12,14,11,22,23], goals [29,28], and returns [37] of opponents during training. Then, the autonomous agent can adapt to different or unseen opponents by using the predictions or representations that are produced by the relevant modules.
> > >
> > > References
> > >
> > > [1-48] References from the original text.
> > >
> > > [49] Tan, M. . "Multi-Agent Reinforcement Learning : Independent vs. Cooperative Agents." Proc. of 10th ICML (1993).
> > >
> > > [50] Shoham, Yoav, and Kevin Leyton-Brown. Multiagent systems: Algorithmic, game-theoretic, and logical foundations. Cambridge University Press, 2008.
> > >
> > > [51] Gintis, Herbert. "Modeling cooperation among self-interested agents: a critique." The journal of socio-economics 33.6 (2004): 695-714.
> > >
> > > [52] Melo FS, Sardinha A. Ad hoc teamwork by learning teammates’ task. Autonomous Agents and Multi-Agent Systems. 2016 Mar;30:175-219.
> > >
> > > [53] Chen S, Andrejczuk E, Irissappane AA, Zhang J. ATSIS: achieving the ad hoc teamwork by sub-task inference and selection. InProceedings of the 28th International Joint Conference on Artificial Intelligence 2019 Aug 10 (pp. 172-179).
> > >
> > > [54] Stone P, Kaminka G, Kraus S, Rosenschein J. Ad hoc autonomous agent teams: Collaboration without pre-coordination. InProceedings of the AAAI Conference on Artificial Intelligence 2010 Jul 5 (Vol. 24, No. 1, pp. 1504-1509).
> > >
> > > [55] Russell, Stuart J., and Peter Norvig. Artificial intelligence: a modern approach. Pearson, 2016.
> > >
> > > [56] Nash, John F. "Non-cooperative games." (1950).
> > >
> > > [57] Nashed, Samer, and Shlomo Zilberstein. "A survey of opponent modeling in adversarial domains." Journal of Artificial Intelligence Research 73 (2022): 277-327.

---

> > > ### Author Response · Authors · 2024-08-12
> > > **Response to reviewer fQcX (Part Ⅱ)**
> > >
> > > Sec 2 Para 6:
> > >
> > > Plan Recognition[58] involves understanding and predicting hidden aspects of an observed entity's trajectory, such as its goals, plans, and underlying policies. Among its key methods are inverse planning [59,60], which emphasizes deducing the decision-making process, and goal recognition [61], which focuses on predicting the ultimate goal or desired final state. DUP[62] approaches plan recognition by using distributed representations of actions to discover plans not found in existing plan libraries. Some works focus on improving the ability of goal recognition instead of RL policies[63,64]. PRP[65] uses model-based algorithm to handle unreliable observations and recognize plans. Unlike existing plan recognition methods, our method aims to enhance policy by opponent modelling, specifically within multi-agent scenarios where no prior knowledge is available.
> > >
> > > References
> > >
> > > [58] Carberry, Sandra. "Techniques for plan recognition." User modeling and user-adapted interaction 11 (2001): 31-48.
> > >
> > > [59] Zhi-Xuan T, Mann J, Silver T, Tenenbaum J, Mansinghka V. Online bayesian goal inference for boundedly rational planning agents. Advances in neural information processing systems. 2020;33:19238-50
> > >
> > > [60] Ying L, Zhi-Xuan T, Mansinghka V, Tenenbaum JB. Inferring the goals of communicating agents from actions and instructions. InProceedings of the AAAI Symposium Series 2023 (Vol. 2, No. 1, pp. 26-33).
> > >
> > > [61] Blaylock, N., Allen, J., 2006. Fast hierarchical goal schema recognition. In: Proceedings of the 21st AAAI National Conference on Artificial Intelligence. pp. 796–801.
> > >
> > > [62] Tian, X., Zhuo, H., Kambhampati, S., 2016. Discovering underlying plans based on distributed representations of actions. In: Proceedings of the 15th International Conference on Autonomous Agents and Multiagent Systems. pp. 1135–1143.
> > >
> > > [63] Vered, M., Kaminka, G., 2017. Heuristic online goal recognition in continuous domains. In: Proceedings of the 26th International Joint Conference on Artificial Intelligence. pp. 4447–4454.
> > >
> > > [64] Shvo M, McIlraith SA. Active goal recognition. InProceedings of the AAAI Conference on Artificial Intelligence 2020 Apr 3 (Vol. 34, No. 06, pp. 9957-9966).
> > >
> > > [65] Sohrabi, S., Riabov, A., Udrea, O., 2016. Plan recognition as planning revisited. In: Proceedings of the 25th International Joint Conference on Artificial Intelligence. pp. 3258–3264.
> > >
> > > Other minor modifications are not listed in detail here, such as the use of symbols $\pi^o, a^o \rightarrow \pi^{-i}, a^{-i}$.

---

> > > ### Author Response · Authors · 2024-08-14
> > > **Response to reviewer fQcX**
> > >
> > > Has my latest response addressed your concerns? If you have any further questions, please let me know. Your feedback is very important to us. Thank you again for your review.

---

> > > ### Author Response · Authors · 2024-08-14
> > > **Response to reviewer fQcX**
> > >
> > > We apologize for not receiving your feedback before the discussion ended. We'd like to share our responses to your concerns once more.
> > >
> > > **Originality:**
> > >
> > > Our method introduces a new framework in opponent modeling. While modeling opponent' subgoal may seem intuitive, it differs significantly from previous methods and from works in plan recognition, inverse planning, and goal recognition.
> > >
> > > **Quality:**
> > >
> > > We've addressed the potential ambiguities you pointed out and enriched the references to relevant literature. Details are in our second response.
> > >
> > > We hope this clarifies our work, and thank you again for your thoughtful review.

---

> > > > ### Comment · Reviewer_fQcX · 2024-08-14
> > > >
> > > > Thank you for your detailed response. I am updating my score to a 5.

---

### Official Review · Reviewer_gEZB · 2024-07-14

**Soundness:** 3
**Presentation:** 3
**Contribution:** 3
**Rating:** 6
**Confidence:** 4

**Summary:**

This paper introduces a multi-agent reinforcement learning algorithm focused on opponent modelling through subgoal inference, termed Opponent Modelling based on Subgoal Inference (OMG). Unlike traditional models that predict immediate actions of opponents, OMG leverages historical trajectories to infer an opponent's future subgoals. This method is designed to enhance generalisation across various unseen opponent policies and includes distinct mechanisms for cooperative and general-sum games. The authors report that their approach outperforms traditional action prediction models in several multi-agent benchmarks.

**Strengths:**

1. The approach of predicting subgoals instead of immediate actions may lead to more robust multi-agent systems. Predicting an opponent's short-term actions tends to provide limited information compared to understanding their long-term strategies.
2. The paper is well-structured, with a clear exposition of the problem, detailed methodology, and coherent discussion of results. Diagrams and formulas are appropriately used to aid understanding.
3. By focusing on subgoals, OMG potentially offers a more scalable and generalizable approach to opponent modelling in multi-agent systems, which could be beneficial for complex scenarios.

**Weaknesses:**

1. “The subgoal prior model, denoted as $p_{\psi}$, is a pre-trained variational autoencoder (VAE) using the states previously collected in the environment”: What policy was used to collect these states to pre-train the prior model? This policy will have a huge impact on the coverage of the state space, and hence the subgoal inference model and final policy of OMG.

2. I have concerns about the generalizability and scalability of this approach. The authors test their approach on SMAC as a complex domain with one easy map (8m) and one medium map (3s_vs_5z). A number of SMAC maps have been shown to be solvable by independent learning methods such as IQL [1] and IPPO [2]. Can the authors showcase performance on hard SMAC maps such as 6h_vs_8z, corridor, 27m_vs_30m which requires learning more complex strategies and have a higher number of agents?

3. Experimental Concerns:

    3a. Why do the authors use different baselines for each environment i.e. D3QN, PPO, and IQL for Foraging, Predator-Prey, and SMAC respectively? Each of these baselines have been used for these environments. [4]

    3b. What variant of PPO was used for the Predator-Prey environment? Both IPPO [2] with individual agent critics and MAPPO [3] with centralised critics have shown to exhibit strong performance in a lot of cooperative benchmarks including predator-prey.

    3c. It would be helpful to assess the impact of the approach if the authors can compare their approach to state-of-the-art CTDE techniques in MARL i.e. QMIX.


[1] QMIX: Monotonic Value Function Factorisation for Deep Multi-Agent Reinforcement Learning

[2] Is Independent Learning All You Need in the StarCraft Multi-Agent Challenge?

[3] The Surprising Effectiveness of PPO in Cooperative, Multi-Agent Games

[4] Benchmarking Multi-Agent Deep Reinforcement Learning Algorithms in Cooperative Tasks

**Questions:**

1. Could the authors provide more insight into how the subgoal predictor's performance might degrade or improve when scaling to environments with significantly more complex or numerous agents?
2. Are there specific types of multi-agent environments where OMG might not perform as expected? What limitations in the model's assumptions should be considered?
3. How does the model handle highly dynamic environments where the opponent's strategies evolve more rapidly than the model's training updates?
4. Please see weaknesses.

**Limitations:**

While the authors have discussed the technical validation of their model, there could be more emphasis on practical limitations, such as computational overhead and real-time applicability in fast-paced environments.

---

> ### Author Rebuttal · Authors · 2024-08-07
>
> Thank you for acknowledging our novel contributions as well as raising valuable questions.
>
> > ''The subgoal prior model, denoted as $p_\psi$, is a pre-trained variational autoencoder (VAE) using the states previously collected in the environment'': What policy was used to collect these states to pre-train the prior model? This policy will have a huge impact on the coverage of the state space, and hence the subgoal inference model and final policy of OMG.
>
> Training OMG requires a set of policies as the training set. During the preparation of the training set, states are collected to train the prior model. The policies used in the training set are described in Appendix B, lines 494-499. As you noted, $p_\psi$, serving as an encoder, must cover the state space as thoroughly as possible. However, it does not limit the policies used to collect states, allowing for an expanded range of state collection in practice.
>
> > I have concerns about the generalizability and scalability of this approach. The authors test their approach on SMAC as a complex domain with one easy map (8m) and one medium map (3s_vs_5z). A number of SMAC maps have been shown to be solvable by independent learning methods such as IQL [1] and IPPO [2]. Can the authors showcase performance on hard SMAC maps such as 6h_vs_8z, corridor, 27m_vs_30m which requires learning more complex strategies and have a higher number of agents?
>
> We tested on 6h_vs_8z, and both QMIX and our method (OMG-optim + 5 QMIX agent) had a win rate of 0%.
> |   | QMIX | OMG-optim + 5 homologue |
> |:-----:|:-----:|:-----:|
> | win rate | 0.0% | 0.0% |
>
> The ability to predict an opponent's actions or subgoals is not the only factor that determines performance, especially in scenarios requiring close cooperation. We are testing on 27m_vs_30m and will provide the results later. This may be challenging due to the persistent scalability issues in opponent modeling.
>
> > 3a. Why do the authors use different baselines for each environment i.e. D3QN, PPO, and IQL for Foraging, Predator-Prey, and SMAC respectively? Each of these baselines have been used for these environments. [4]
>
> The core of OMG is opponent modeling method, which does not restrict the type of underlying RL method. Therefore, we followed the original environment's setup.
>
> > 3b. What variant of PPO was used for the Predator-Prey environment? Both IPPO [2] with individual agent critics and MAPPO [3] with centralised critics have shown to exhibit strong performance in a lot of cooperative benchmarks including predator-prey.
>
> We used IPPO. The centralized critics are not applicable to the setup of this study. Because, the motivation of this paper is to address the autonomous agent through opponent modeling, rather than training a group of agents for some task.
>
> > 3c. It would be helpful to assess the impact of the approach if the authors can compare their approach to state-of-the-art CTDE techniques in MARL i.e. QMIX.
>
> We has conducted tests where QMIX acts as the agent with opponents of test set in **Appendix C**.
>
> The training paradigm employed by QMIX leads to a lack of generalization for different opponents. Using the same training methodology for QMIX as OMG leads to a degradation in IQL, which already serves as one of the existing baselines. The test results indicate that opponents trained using different methods and seeds are not homogeneous, which poses challenges for cooperation.
>
> > Could the authors provide more insight into how the subgoal predictor's performance might degrade or improve when scaling to environments with significantly more complex or numerous agents?
>
> When modeling opponents, the difficulty of prediction usually escalates in environments with more complex or numerous agents, as the prediction space expands. From a scalability standpoint, OMG offers certain advantages over traditional action prediction methods. For example, if the state space of the environment excludes agent information, OMG's prediction space does not significantly increase with the number of agents.
>
> > Are there specific types of multi-agent environments where OMG might not perform as expected? What limitations in the model's assumptions should be considered?
>
> We believe it is essential for opponents in the environment to have clear goals. An opponent with a purposeless, random policy is challenging to model, even though it might not directly impact the agent's rewards.
>
> > How does the model handle highly dynamic environments where the opponent's strategies evolve more rapidly than the model's training updates?
>
> OMG does not account for scenarios where opponents rapidly evolve during interactions. Addressing such situations requires incorporating planning ideas as discussed in [1, 2], which is left for future work. Currently, OMG addresses this by ensuring that the training set includes a diverse range of opponent policies, allowing it to generalize well even when opponents rapidly evolve during testing.
>
> [1] Zhi-Xuan T, Mann J, Silver T, Tenenbaum J, Mansinghka V. Online bayesian goal inference for boundedly rational planning agents. Advances in neural information processing systems. 2020;33:19238-50
>
> [2] Xiaopeng Yu, Jiechuan Jiang, Wanpeng Zhang, Haobin Jiang, and Zongqing Lu. Model-based opponent modeling. Advances in Neural Information Processing Systems, 35:28208–28221, 2022.

---

> > ### Author Response · Authors · 2024-08-12
> > **Response to reviewer gEZB**
> >
> > The test performance on 27m_vs_30m as follows:
> >
> > |   | OMG-optim | LIAM | Naive OM |
> > |:-----:|:-----:|:-----:|:-----:|
> > | win rate | 47.4% ± 14.1% | 39.0% ± 9.3% | 38.6% ± 15.0% |
> >
> > The homogeneous agents collaborating with the baselines are trained using the QMIX algorithm, achieving a win rate of 49%. OMG outperforms these baselines and reaches a win rate close to that of the well-trained QMIX team, demonstrating its effectiveness even in environments with a larger number of opponents.

---

> > > ### Author Response · Authors · 2024-08-13
> > > **To reviewer gEZB**
> > >
> > > Has my response addressed your concerns? If there are any remaining issues, please let me know. If everything is clear, could you consider adjusting the score? Thank you sincerely for your review.

---

> ### Comment · Reviewer_gEZB · 2024-08-13
> **Thanks for the rebuttal**
>
> Thank you for your rebuttal. I appreciate the additional clarifications provided and have decided to update my score, provided the additional experiments are added to the final version of the paper.

---

> > ### Author Response · Authors · 2024-08-13
> > **Response to reviewer gEZB**
> >
> > Thank you sincerely again for your comments and appreciation.

---

### Author Rebuttal · Authors · 2024-08-07

We have added two experiments in the PDF as follows:

* We have increased the number of seeds in the Predator-Prey model from 5 to 10 to reduce experimental error and enhance the reliability of our findings.

* We have incorporated an ablation study on subgoal selection, where OMG-Random, OMG-1s, and OMG-3s represent subgoals selected from the opponent’s future states randomly, at the next step, and at three steps, respectively.

---

### Decision · Program_Chairs · 2024-09-25

**Decision:**

Accept (poster)

**Comment:**

This multi-agent RL paper is on ad hoc teamwork and is built around the idea of inferring co-player subgoals. Reviewers mostly appreciated the paper's clear presentation, though one identified specific presentation issues which they said should be fixed in the final version. I also had a minor nitpick with the presentation too in that I noticed that the following reference, while deeply relevant to the paper, was missing from the discussion:

Vezhnevets, Alexander, et al. "Options as responses: Grounding behavioural hierarchies in multi-agent reinforcement learning." International Conference on Machine Learning. PMLR, 2020.

But presentation aside, no one raised very significant concerns with the novelty of the approach. Moreover, the model and its evaluation were seen as technically sound, and the reviewers appreciated the author's detailed responses during the rebuttal period.